


# WRF-Chem simulated surface ozone over South Asia during the pre-monsoon: Effects of emission inventories and chemical mechanisms

**Amit Sharma[1,2,*], Narendra Ojha[2,*], Andrea Pozzer[2], Kathleen A. Mar[3], Gufran Beig[4], Jos Lelieveld[2,5], and Sachin S. Gunthe[1]**

[1]Department of Civil Engineering, Indian Institute of Technology Madras, Chennai, India
[2]Atmospheric Chemistry Department, Max Planck Institute for Chemistry, Mainz, Germany
[3]Institute for Advanced Sustainability Studies, Potsdam, Germany
[4]Indian Institute for Tropical Meteorology, Pune, India
[5]Energy, Environment and Water Research Center, The Cyprus Institute, Nicosia, Cyprus

[*]*Correspondence to*: Amit Sharma (amit.iit87@gmail.com) and Narendra Ojha (narendra.ojha@mpic.de)

**Abstract**

We evaluate numerical simulations of surface ozone mixing ratios over the South Asian region during the pre-monsoon season employing three different emission inventories (EDGAR-HTAP, INTEX-B, and SEAC4RS) in the WRF-Chem model with the RADM2 chemical mechanism. Evaluation of modelled ozone and its diurnal variability, using data from a network of 18 monitoring stations across South Asia, show the model ability to reproduce the clean, rural and polluted urban environments over this region. In contrast to the diurnal average, the modelled ozone mixing ratios during the noontime i.e. hours of intense photochemistry (1130-1630 h Indian Standard Time or IST) are found to differ among the three inventories. This suggests that evaluations of the modelled ozone limited to 24-h average are insufficient to comprehend the uncertainties associated with ozone build-up. HTAP generally shows 10-30 ppbv higher noontime ozone mixing ratios than SEAC4RS and INTEX-B, especially over the north-west Indo-Gangetic Plain (IGP), central India and southern India. Further, the model performance shows strong spatial heterogeneity, with SEAC4RS leading to better agreement with observations over east and south India, whereas HTAP performs better over north and central India, and INTEX-B over west India. The Normalized Mean Bias (NMB in %) in the noontime ozone over the entire South Asia is found to be lowest for the SEAC4RS (~11%), followed by INTEX-B (~12.5%) and HTAP (~22%). The HTAP simulation repeated with the alternative MOZART chemical mechanism showed even more strongly enhanced surface ozone mixing ratios (noontime NMB=36.5%) due to vertical mixing of enhanced ozone that has been produced aloft. The SEAC4RS inventory with the RADM2 chemical mechanism is found to be the most successful overall among the configurations evaluated here in simulating ozone air quality over South Asia. Our study indicates the need to also evaluate the $O_3$ precursors across a network of stations to further reduce uncertainties in modelled ozone.



## 1. Introduction

Tropospheric ozone plays central roles in atmospheric chemistry, air quality and climate change. Unlike primary pollutants, which are emitted directly, tropospheric ozone forms photochemically involving precursors such as carbon monoxide (CO), volatile organic compounds (VOCs) and oxides of nitrogen ($NO_x$), supplemented by transport from the stratosphere (e.g. Crutzen, 1974; Atkinson, 2000; Monks et al., 2015). It can be transported over long distances resulting in enhanced concentrations even in areas located remote from the sources of precursors (Cox et al., 1975). The photochemical production of ozone and its impacts on agricultural crops and human health are especially pronounced near the surface. Numerous studies have shown that elevated surface ozone levels significantly reduce crop yields (e. g.; Krupa et al., 1998; Emberson et al., 2009; Ainsworth et al., 2012; Wilkinson et al., 2012), in addition to adverse human health effects that cause premature mortality (e.g., Bell et al., 2004; Jerrett et al., 2009; Anenberg et. al., 2010; Lelieveld et al., 2015).

An accurate representation of anthropogenic emissions of ozone precursors is essential to understand the photochemical production of ozone and support policy making. While anthropogenic emissions have been nearly stable or decreasing over northern America and Europe (e. g. Yoon and Pozzer, 2014), there has been substantial enhancement over the East and South Asian regions in recent decades (e. g. Akimoto, 2003; Ohara et al., 2007, Logan et al., 2012; Gurjar et al., 2016). The number of premature mortalities per year due to outdoor air pollution is anticipated to double by the year 2050 as compared to the year 2010 in a business-as-usual scenario, predominantly in Asia (Lelieveld et al., 2015). The multi-pollutant index over all populated regions in the northern hemisphere shows a general increase, with South Asia being the major hotspot of deteriorating air quality (Pozzer et al., 2012).

The growth of anthropogenic emissions over the South Asian region has regional implications, and is also predicted to influence air quality on a hemispheric scale (Lelieveld and Dentener, 2000). It was shown that the anthropogenic emissions and their subsequent photochemical degradation over South Asia influence air quality over the Himalayas (e.g. Ojha et al., 2012; Sarangi et al., 2014) and the Tibetan Plateau (Lüthi et al., 2015) as well as the marine environment downwind of India (e.g. Lawrence and Lelieveld, 2010). Additionally, the prevailing synoptic scale weather patterns make this region highly conducive to long-range export of pollutants (e.g. Lelieveld et al., 2002; Lawrence et al., 2003; Ojha et al., 2014; Zanis et al., 2014). Therefore, the accurate estimation of anthropogenic emissions over South Asia and their representation in chemical transport models are essential to quantify the effects on regional as well as global air quality.

The Weather Research and Forecasting model with Chemistry (WRF-Chem) (Grell et al., 2005; Fast et al., 2006), a regional simulation system, has been popular for use over the South Asian region in numerous recent studies to simulate the meteorology and spatio-temporal distribution of ozone and related trace gases (e. g. Kumar et al., 2012a, 2012b; Michael et al., 2013; Gupta et al., 2015; Jena et al., 2015; Ansari et al., 2016; Ojha et al., 2016; Girach et al., 2016). WRF-Chem simulations at higher spatial resolution employing regional emission inventories have been shown to better reproduce the observed spatial and temporal heterogeneities in ozone over this region as compared to the global models (e.g. Kumar et al., 2012b; Ojha et al., 2016). However, an evaluation of modelled ozone based on data from a network of stations across South Asia is imperative considering very large spatio-temporal heterogeneity in the distribution of ozone over this region (e.g. Kumar et al., 2010; Ojha et al., 2012; Kumar et al., 2012b) mainly resulting from heterogeneous precursor sources. WRF-Chem simulated ozone





distributions have also been utilized to assess the losses in crop yields, and it was suggested that the estimated crop
losses would be sufficient to feed about 94 million people living below the poverty line in this region (Ghude et
al., 2014). Further, WRF-Chem has been used to estimate that premature mortality in India caused by chronic
obstructive pulmonary disease (COPD) due to surface $O_3$ exposure was ~12,000 people in the year 2011 (Ghude et
al., 2016). Despite these applications, there is room for improvement in modeled concentrations as some limited
studies evaluating ozone on diurnal scales revealed a significant overestimation of noontime ozone e.g. by as much
as 20 ppbv in Kanpur (Michael et al., 2013) and 30 ppbv in Delhi (Gupta and Mohan, 2015).
Using WRF-Chem, Amnuaylojaroen et al. (2014) showed that over continental southeast Asia surface ozone
mixing ratios vary little (~4.5%) among simulations employing different emission inventories. A recent study by
Mar et al. (2016) highlighted the dependence of WRF-Chem predicted ozone air quality (over Europe) on the
chosen chemical mechanism. These results indicate the need for evaluating the effects of emission inventories and
chemical mechanisms on the model performance using a network of stations across South Asia, which has not
been carried out thus far. The main objectives of the present study are:
(a) To evaluate WRF-Chem simulated ozone over South Asia, including the diurnal cycle, against recent in situ
measurements from a network of stations;
(b) To inter-compare model simulated $O_3$ among different emission inventories;
(c) To inter-compare model simulated $O_3$ between two extensively used chemical mechanisms (MOZART and
RADM2) with the same emission inventory;
(d) To provide recommendations on the model configuration for future studies over stations, sub-regions as well
as the entire South Asian region.

We focus on the pre-monsoon season (March-May) for the study as $O_3$ mixing ratios at the surface are generally
the highest over most of South Asia during this period (Jain et al., 2005; Debaje et al., 2006; Reddy et al., 2010;
Ojha et al., 2012; Gaur et al., 2014; Renuka et al., 2014; Bhuyan et al., 2014; Sarangi et al., 2014; Yadav et al.,
2014; Sarkar et al., 2015). This is because photochemistry over South Asia is most intense during this season
caused by the combined effects of high pollution loading, biomass-burning emissions and a lack of precipitation.
Section 2 presents the model description, including physics and chemistry options, emission inputs and the
observational data. Model evaluation focussing on the effects of different emission inventories on ozone is
presented in section 3. The inter-comparison between the RADM2 and MOZART chemical mechanism is
discussed in section 4. For the list of symbols and acronyms used in this paper are listed in Table 1. The sub-
regional and South Asian domain evaluation and recommendations on model configuration are provided in section
5, followed by the summary and conclusions drawn from the study in section 6.
**2. Methodology**
**2.1. WRF-Chem**
In this study we use the Weather Research and Forecasting model coupled with chemistry (WRF-Chem version
3.5.1), which is an online mesoscale model capable of simulating meteorological and chemical processes
simultaneously (Grell et al., 2005; Fast et al., 2006). The model domain (Fig. 1) is defined on a mercator
projection and is centred at $22^0$ N, $83^0$ E with 274 and 352 grid points in the east-west and north-south directions,
respectively, at the horizontal resolution of 12 km x 12 km. The land use data is incorporated from the US



Geological Survey (USGS) based on 24 land use categories. The ERA-interim reanalysis dataset from ECMWF
(http://www.ecmwf.int/en/research/climate-reanalysis/browse-reanalysis-datasets), archived at the horizontal
resolution of about $0.7^o$ and temporal resolution of 6 hours, is used to provide the initial and lateral boundary
conditions for the meteorological calculations. All simulations in the study have been conducted for the period:
$26^{th}$ February – $31^{st}$ May, 2013 at a time step of 72 s. The model output is stored every hour for analysis. The first
three days of model output have been discarded as model spin up.
Radiative transfer in the model has been represented using the Rapid Radiative Transfer Model (RRTM) longwave
scheme (Mlawer, 1997) and the Goddard shortwave scheme (Chou and Suarez, 1994). Surface physics is
parameterized using the Unified Noah land surface model (Tewari et al., 2004) along with eta similarity option
(Monin and Obukhov, 1954; Janjic, 1994, 1996), and the planetary boundary layer (PBL) is based on the Mellor-
Yamada-Janjic (MYJ) scheme (Mellor and Yamada, 1982; Janjic, 2002). The cloud microphysics is represented
by the Lin et al. scheme (Lin et. al., 1983), and cumulus convection is parameterized using the Grell 3D Ensemble
Scheme (Grell, 1993; Grell and Devenyi, 2002). Four-dimensional data assimilation (FDDA) is incorporated for
nudging to limit the drift in the model simulated meteorology from the ERA-interim reanalysis (Stauffer and
Seaman, 1990; Liu et al. 2008). Horizontal winds are nudged at all vertical levels, whereas temperature and water
vapour mixing ratios are nudged above the PBL (Stauffer et al. 1990, 1991). The nudging coefficients for
temperature and horizontal winds are set as $3 \times 10^{-4}$ $s^{-1}$ whereas it is set as $10^{-5}$ $s^{-1}$ for water vapour mixing ratio
(Otte, 2008).
This study utilizes two different chemical mechanisms, the Regional Acid Deposition Model - $2^{nd}$ generation
(RADM2) (Stockwell et al., 1990), and the Model for Ozone and Related Chemical Tracers-version 4 (MOZART-
4) (Emmons et al., 2010). RADM2 chemistry includes 63 chemical species participating in 136 gas phase and 21
photolysis reactions. MOZART chemistry includes 81 chemical species participating in 159 gas phase and 38
photolysis reactions. Aerosols are represented using the Modal Aerosol Dynamics Model for Europe/ Secondary
Organic Aerosol Model (MADE/ SORGAM) (Ackermann et al., 1998; Schell et al., 2001) with RADM2 and
Global Ozone Chemistry Aerosol Radiation and Transport (GOCART) (Chin et al., 2000) with MOZART. The
photolysis rates are calculated using Fast-J photolysis scheme (Wild et al., 2000) in RADM2 simulations and
Madronich F-TUV scheme in MOZART simulation. The Madronich F-TUV photolysis scheme uses
climatological $O_3$ and $O_2$ overhead columns. The treatment of dry deposition process also differs between RADM2
and MOZART owing to differences in Henry's Law coefficients and diffusion coefficients. The chemical initial
and lateral boundary conditions are provided from 6 hourly fields from the Model for Ozone and Related
Chemical Tracers (MOZART-4/GEOS5) (http://www.acom.ucar.edu/wrf-chem/mozart.shtml).
**2.2. Emission inputs**
This study utilizes three different inventories for the anthropogenic emissions: HTAP, INTEX-B and the
SEAC4RS, which are briefly described here. The Hemispheric Transport of Air Pollution (HTAP) inventory
(Janssens-Maenhout et al., 2015) for anthropogenic emissions (http://edgar.jrc.ec.europa.eu/htap_v2
/index.php?SECURE=_123) available for the year 2010 has been used. The HTAP inventory has been developed
by complementing various regional emissions with EDGAR data, in which Asian region including India is
represented by the Model Intercomparison study for Asia (MICS-Asia) inventory, which is at a horizontal
resolution of $0.25^o$ x $0.25^o$ (Carmichael et al., 2008). The resultant global inventory is re-gridded at the spatial



resolution of 0.1° x 0.1° and temporal resolution of 1 month. HTAP includes emissions of CO, $NO_x$, $SO_2$,
NMVOCs, PM, BC and OC from power, industry, residential, agriculture, ground transport and shipping sectors.
The Intercontinental Chemical Transport Experiment-Phase B (INTEX-B) inventory (Zhang et al., 2009),
developed to support the INTEX-B field campaign by the National Aeronautics and Space Administration
(NASA) in spring 2006, is the second inventory used in this study. It provides total emissions for year 2006 at a
horizontal resolution of 0.5° x 0.5°. The emission sectors include power generation, industry, residential and
transportation. The Southeast Asia Composition, Cloud, Climate Coupling Regional Study (SEAC4RS) inventory
(Lu and Streets, 2012), prepared for the NASA SEAC4RS field campaign, is the third inventory used in this study.
It provides total emissions for the year 2012 at a spatial resolution of 0.1° x 0.1°. The SEAC4RS and INTEX-B did
not cover regions in the north western part of the domain, and therefore we complemented this region (longitude <
75°E and latitude > 25°N) by HTAP emission data. The emissions of CO, NMVOCs and $NO_x$ emissions among
the three emission inventories, as included in the simulations, are shown in Fig. 2. Table 2 provides estimates of
total emissions over different regions (as defined in Fig.1) from the three inventories. The emissions from biomass
burning are included using the Fire Inventory from NCAR (FINN) version 1.0 (Wiedinmyer et al., 2011). Model
of Emissions of Gases and Aerosols from Nature (MEGAN) is used to include the biogenic emissions (Guenther et
al., 2006) in the model.
The HTAP inventory is available at monthly temporal resolution while INTEX-B and SEAC4RS are available as
annual averages; however, seasonal variability in anthropogenic emissions may not have a major effect in this
study as we focus here on spring (pre-monsoon), for which monthly emissions are similar to the annual mean
(seasonal factor close to unity) (Supplementary material - Fig. S1; also see Fig. 2b in Kumar et al., 2012b).
Nevertheless, seasonal influence during spring is strongest for biomass-burning emissions, which have been
accounted for. The emissions from all inventories were injected in the lowest model layer. The diurnal profiles of
the anthropogenic emissions of ozone precursors, specific to South Asia are not available. A sensitivity simulation
implementing the diurnal emission profile available for Europe (Mar et al., 2016 and references therein) showed a
little impact on predicted noontime ozone over South Asia (Supplementary material – Fig S2).
**2.3.   Simulations**
We have conducted 4 different numerical simulations as summarized in Table 3 and briefly described here. Three
simulations correspond to three different emission inventories HTAP, INTEX-B and SEAC4RS for the
anthropogenic emissions of ozone precursors, employing the RADM2 chemical mechanism. These simulations are
named HTAP-RADM2, INTEX-RADM2 and S4RS-RADM2 respectively. The emissions of aerosols have been
kept same (HTAP) among these three simulations and aerosol-radiation feedback has been switched off to
specifically identify the effects of emissions of $O_3$ precursors on modelled ozone. An additional simulation HTAP-
MOZ has been conducted to investigate the sensitivity of ozone to the employed chemical mechanism (MOZART
vs RADM2) by keeping the emissions fixed to HTAP.
**2.4. Observational dataset**
Surface ozone data is acquired from various studies and sources, as given in Table 4. In general, surface $O_3$
measurements over these stations have been conducted using the well-known technique of UV light absorption by
ozone molecules at about 254 nm, making use of Beer-Lambert's Law. The accuracy of these measurements is
reported to be about 5% (Kleinmann et al., 1994). The response time of such instruments is about 20 s and





instruments have a lower detection limit of 1 ppbv (Ojha et al., 2012). Here we have used the hourly and monthly
average data for the model evaluation. The details of instruments and calibrations at individual stations can be
found in the references given in the Table 4.
As simultaneous measurements at different stations are very sparse over South Asia, the model evaluation has
often to be conducted using observations of the same season/month of a different year (e. g. Kumar et al., 2012b;
Kumar et al., 2015; Ojha et al., 2016). However, to minimize the effect of temporal differences we preferentially
used measurements of recent years i.e. the observations at ~83% of the stations used in this study are of the period:
2009-2013. For four stations: Delhi (north India), Jabalpur (central India), Pune (west India) and Thumba (south
India), the observations and simulations are for the same year (2013). The observations at three stations have been
collected in previous periods (2004 or before). Finally, we investigated the effects of temporal differences on the
results and model biases presented here by conducting another simulation for a different year (2010)
(Supplementary material, Fig. S3).
**3. Effects of emission inventories**
**3.1. Spatial distribution of Ozone**
The spatial distribution of WRF-Chem simulated 24-h monthly average ozone during April is shown in Fig. 3a
(upper panel) for the three different emission inventories (HTAP, INTEX, and SEAC4RS). Generally the months
of March and May are marked with seasonal transition from winter to summer and summer to monsoon
respectively. Hence, the month of April is chosen to represent the pre-monsoon season as it is not influenced by
these seasonal transitions, and the observational data is available for a maximum number of stations during this
month for the comparison. The 24-h average ozone mixing ratios are found to be 40-55 ppbv over most of the
Indian subcontinent for all the three inventories. Model simulated ozone levels over the coastal regions are also
similar (30-40 ppbv) among the three inventories. The highest ozone mixing ratios (55 ppbv and higher) predicted
in the South Asian region are found over northern India and the Tibetan Plateau. The WRF-Chem simulated
spatial distributions of average ozone shown here are in agreement with a previous evaluation study over South
Asia (Kumar et al., 2012b). Further, it is found that qualitatively as well as quantitatively the HTAP, INTEX-B
and SEAC4RS lead to very similar distributions of 24-h average ozone over most of the South Asian region. The
24h monthly average ozone from observations is superimposed on the model results in Fig. 3a for comparison.
WRF-Chem simulated distributions of average $O_3$ are in general agreement with the observational data (Fig. 3a),
except at a few stations near coasts (e. g. Kannur and Thumba) and in complex terrain (Pantnagar and Dibrugarh).
In contrast to the distribution of 24-h average $O_3$, the noontime (1130-1630 IST) $O_3$ mixing ratios over continental
South Asia exhibit significant differences among the three emission inventories (Fig. 3b). HTAP clearly leads to
higher noontime $O_3$ mixing ratios, the difference being up to 10 ppbv over the Indo-Gangetic plain (IGP), 20 ppbv
over Central India, 30 ppbv over Southern India, compared to INTEX-B and SEAC4RS. The mean bias (MB)
(model-observation) for 24-h and noontime average ozone at individual stations is provided in the supplementary
material - Table S1 and S2.

The net photochemical $O_3$ production rate (ppbv h$^{-1}$) from sunrise to noontime (0630-1230 IST), when most of the
photochemical build-up of ozone takes place and leading to its peak noontime mixing ratio, has been calculated
utilizing the chemical tendencies in WRF-Chem (Barth et al., 2012; Girach et al., 2016). A comparison of monthly
average $O_3$ production rates among the three inventories is shown in Fig. 4. As seen also from the $O_3$ mixing ratios

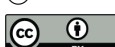



(Fig. 3b), the HTAP emissions result in faster $O_3$ production (~9 ppbv h$^{-1}$) throughout the IGP region. The highest
$O_3$ production rates for INTEX-B and SEAC4RS inventories are simulated only in the East Indian regions
including the eastern parts of the IGP. It is noted that the rate of $O_3$ production is lower (4-8 ppbv h$^{-1}$) over most of
the south-western IGP for the INTEX-B and SEAC4RS inventories. Differences are also found over the southern
Indian region with stronger ozone production in HTAP, followed by INTEX-B and SEAC4RS.

Figure 5 provides insight into the spatial distribution of $O_3$ production regimes estimated through the $CH_2O/NO_y$
ratio (Geng et al., 2007; Kumar et al. 2012b) calculated during 0630 – 1230 IST, to help explain the differences in
modelled ozone mixing ratios among the three simulations. The spatial distribution of regimes in all simulations is
largely consistent with the findings of Kumar et al. (2012b) although the latter performed the analysis for
afternoon hours (1130 – 1430 IST). The S4RS-RADM2 simulation predicts the entire IGP to be VOC sensitive
whereas in HTAP-RADM2 and INTEX-RADM2 simulations though the northwest IGP and eastern IGP are VOC
sensitive, the central IGP is mostly $NO_x$ limited. The coastal regions are also predicted to be VOC limited in all the
three simulations. With the north-western IGP being VOC limited in all simulations, the noontime ozone mixing
ratios are found to be higher in this region in HTAP-RADM2 simulation because of high NMVOC emissions in
HTAP inventory as evident from figure 2 and table 2. Similar differences are also apparent in southern India.

In summary, these results show similar 24-h average ozone distributions but large differences in the ozone build-
up until noon. The net photochemical ozone production in the morning hours (0630-1230) is shown to be sensitive
to the different inventories over this region, which is attributed to differences in total $NO_x$ and/or NMVOC
emissions. We therefore suggest that a focus on 24-h averages only would be insufficient to evaluate the ozone
budget and implications for human health and crops. Next we compare the modeled diurnal ozone variations from
three inventories with in situ measurements over 18 stations across the South Asia.

**3.2. Diurnal variation**
A comparison of WRF-Chem simulated diurnal ozone variability with recent in situ measurements over a network
of 18 stations in the South Asian region is shown in Fig. 6. WRF-Chem is found to successfully reproduce the
characteristic diurnal ozone patterns observed over the urban (e.g. Mohali, Delhi, Kanpur, Ahmedabad,
Bhubaneswar and Pune) and rural (e.g. Joharapur, Anantpur, Gadanki) stations, indicating strong ozone build-up
from sunrise to noontime and the predominance of chemical titration (by NO) and deposition losses during the
night. In general, WRF-Chem captures the daily amplitude of $O_3$ changes at relatively cleaner and high altitude
stations, typically showing less pronounced diurnal variability, such as Nainital in the Himalayas and Mt. Abu in
the Aravalli mountain range, although with differences in timing when model and observations attain minimum
ozone mixing ratios, thus leading to relatively low correlation coefficient (see later in the text). For example,
modelled diurnal amplitudes at Nainital are estimated to be ~19.2 ppbv (HTAP-RADM2), ~17.5 ppbv (INTEX-
RADM2) and ~17.9 ppbv (S4RS-RADM2) as compared to the observational value of ~15.1 ppbv. The model does
not reproduce the ozone mixing ratios at Pantnagar and Jabalpur except for afternoon peak values. This can be
attributed to the role of complex terrain (presence of the Himalayas near Pantnagar), which cannot be fully
resolved, even at 12 km resolution. Jabalpur is also surrounded by forests, hills and mountains (Sarkar et al.,
2015), and such variability in a small area could impact the accuracy of model predictions. The model typically



overestimates the noontime ozone mixing ratios over several urban (e.g. Kanpur, Ahmedabad, Haldia, Thumba)
and rural stations (e.g. Joharapur, Kannur), which is attributed to the uncertainties in the emissions.
To briefly evaluate the possible effects due to the difference in meteorological year between model and
observations, we repeated the HTAP-RADM2 simulation for a different year (2010) as shown in the
Supplementary material – Fig. S3. The effect of changing the meteorological year in the model simulation is
generally small (mostly within ±3 ppbv in 3 years), except at a few stations in the east (Haldia and Bhubaneswar)
and north (Nainital and Pantnagar). The effect is seen to vary from 4.8 ppbv to 11 ppbv (in 3 years) at these four
stations. These differences are found to be associated with the inter-annual variations in the regional and
transported biomass burning emissions, as seen from MODIS fire counts and MOZART/GEOS5 boundary
conditions (not shown).
The model ability to reproduce diurnal variations at all stations is summarised using a Taylor diagram (Taylor,
2001) in Figure 7. The statistics presented are normalised standard deviation (SD), normalised centred root mean
squared difference (RMSD) and the correlation coefficient. The normalisation of both SD and RMSD is done
using the standard deviation of the respective observational data. The point indicated as 'REF' represents the
observational data against the model results evaluated. WRF-Chem simulations show reasonable agreement with
observations with correlation coefficients generally greater than 0.7 for most sites. The locations such as Nainital,
Mt. Abu and Jabalpur for which r values are lower (0.3-0.7) are associated with unresolved complex terrain, as
mentioned earlier.

### 4. Effects of chemical mechanism (RADM2 vs MOZART)

A recent WRF-Chem evaluation over Europe showed better agreement with in situ measurements when the
MOZART chemical mechanism was employed, compared to RADM2 (Mar et al., 2016). Following up on this,
here we compare modelled ozone mixing ratios obtained with these two extensively used chemical mechanisms
over South Asia: RADM2 (e. g. Kumar et al., 2012b; Michael et al., 2013; Ojha et al., 2016, Girach et al., 2016)
and MOZART (e. g. Ghude et al., 2014; Ghude et al., 2016), keeping the same input emission inventory (HTAP).
Thus, the following sensitivity analysis is aimed at exploring if the use of the more detailed chemical mechanism
of MOZART could improve the model performance.

### 4.1. Spatial distribution of surface O$_3$

The WRF-Chem simulated spatial distributions of 24-h average and noontime average surface ozone are compared
in Fig. 8. The monthly values of the 24-h and noontime ozone mixing ratios from measurements are also shown.
Overall, the average ozone mixing ratios over South Asia are simulated to be higher with the MOZART chemical
mechanism compared to RADM2, which is consistent with the results of Mar et al. (2016) for the European
domain. The 24-h average ozone mixing ratios over India simulated with MOZART chemistry are found to be
higher than those with RADM2 chemistry, especially over the eastern Indian region (~60 ppbv and more for
MOZART compared to ~40-55 ppbv for RADM2). Average ozone levels over the coastal regions are found to be
similar between the two mechanisms (30-40 ppbv). MOZART chemistry also predicts high 24-h average ozone
mixing ratios (55 ppbv and higher) over the Tibetan Plateau region, similar to RADM2. A striking difference
between the two chemical mechanisms is found over the marine regions adjacent to South Asia (Bay of Bengal
and northern Indian Ocean), with MOZART predicting significantly higher 24-h average ozone levels (35-50
ppbv) compared to the RADM2 (25-40 ppbv). A comparison of noontime average ozone distributions between the



two chemical mechanism shows that MOZART predicts higher ozone concentrations than RADM2 over most of
the Indian region by about 5-20 ppbv, except over western India. The differences are up to 20 ppbv and more over
the Southern Indian region, highlighting the impacts of chemical mechanisms on modelled ozone in this region.
The mean bias (MB) values (model-observation) for 24-h and noontime average ozone at individual stations is
provided in the supplementary material - Table S1 and S2.

Figure 9a shows a comparison of the monthly average chemical $O_3$ tendency (ppbv h$^{-1}$) from 0630 to 1230 IST. In
contrast with average $O_3$ mixing ratios, which were found to be higher in HTAP-MOZ, the net $O_3$ production rates
at the surface are higher in HTAP-RADM2 over most of the domain, especially in the IGP and central India. The
net $O_3$ production rates at the surface with HTAP-RADM2 are found to be 6 to 9 ppbv h$^{-1}$ and more over the IGP,
whereas these values are generally lower in HTAP-MOZ (4-8 ppbv h$^{-1}$), except in the north-eastern IGP (>9 ppbv
h$^{-1}$). Fig. 9b shows the sum of the chemical tendency and vertical mixing tendency at the surface for the HTAP-
RADM2 and HTAP-MOZ. Analysis of the vertical mixing tendency revealed that higher surface ozone mixing
ratios in the MOZART simulation are due to mixing with ozone rich air from aloft. In the HTAP-RADM2
simulation, vertical mixing dilutes the effect of strong chemical surface ozone production. Further analysis of
vertical distributions of chemical $O_3$ tendencies reveals stronger photochemical production of ozone aloft with
MOZART compared to RADM2 (Supplementary material-Fig. S4). This leads to higher ozone mixing ratios aloft
in MOZART simulations.
Mar et al. (2016) showed that RADM2 exhibits greater VOC sensitivity than MOZART (i.e., producing higher
changes in ozone given a perturbation in VOC emissions) under noontime summer conditions over Europe. This is
consistent with our findings as well, that the net surface photochemical ozone production is greater for HTAP-
RADM2 than for HTAP-MOZART, given the high VOC emissions in the HTAP inventory. At the surface, the
MOZART mechanism predicts larger areas of VOC-sensitivity (as diagnosed by the $CH_2O/NO_y$ indicator, Figure
10) and lower net photochemical ozone production than RADM2. With increasing altitude, both the HTAP-
RADM2 and HTAP-MOZART simulations show a general increase of $CH_2O/NO_y$ over India, i.e. the chemistry
tends to exhibit increased $NO_x$ sensitivity with increasing height (Supplementary material-Figure S6). At model
levels above the surface, HTAP-MOZART shows greater net photochemical production of ozone than HTAP-
RADM2 (Supplementary material-Figure S4), which is what Mar et al. (2016) have also reported for the surface
$O_3$ over Europe. When these effects are combined, mixing leads to higher surface ozone mixing ratios for HTAP-
MOZART than for HTAP-RADM2. The differences in ozone mixing ratios between the MOZART and RADM2
chemical mechanisms can be attributed to the additional chemical species and reactions, differences in the rate
constants for several inorganic reactions, and photolysis schemes used, with relatively smaller effects from $O_3$ dry
deposition (as in Mar et al. 2016). Our analysis also shows the importance of chemical regime in understanding
differences between the chemical mechanisms, and highlights the significant effects of the employed chemical
mechanism on modelled ozone over South Asia.

### 4.2. Diurnal variation

Figure 11 shows a comparison of WRF-Chem simulated ozone variations on diurnal timescales with recent in situ
measurements over a network of stations across the South Asia for the two chemical mechanisms (MOZART and
RADM2); again with the same emission inventory (HTAP). Qualitatively, both simulations produce very similar
diurnal patterns, however, the absolute $O_3$ mixing ratios are found to differ significantly between the two chemical

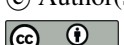

mechanisms. Noontime ozone mixing ratios predicted by MOZART are either significantly higher (at 12 out of 18
stations) or nearly similar (at 6 stations). MOZART-predicted $O_3$ at Dibrugarh, Kanpur, Jabalpur, Bhubaneshwar,
Gadanki and Thumba was found to be higher by ~12 ppbv, 5 ppbv, 8 ppbv, 10 ppbv, 11 ppbv and 12 ppbv,
respectively, compared to RADM2 (Supplementary material, Table S2). Over several urban and rural stations in
India (e.g. Delhi, Ahmedabad, Pune, Kannur and Thumba) MOZART is found to titrate ozone more strongly
during the night while resulting in higher or similar ozone levels around noon. The contrasting comparison
between noon and night time found at these sites suggests that evaluation limited to 24 h averages would not be
sufficient, and that model performance on a diurnal time scale should be considered to assess the photochemical
build-up of $O_3$.

In general, the noontime ozone mixing ratios predicted by RADM2 are found to be in better agreement with in situ
measurements compared to MOZART. The model performance of two chemical mechanisms in reproducing
diurnal variation at all stations is summarised using a Taylor diagram in Fig. 12. Both chemical mechanisms show
reasonably good agreement (r > 0.7) at most of the sites, except two stations associated with highly complex
terrain (Nainital and Mt. Abu). On the Taylor diagram, most of the HTAP-RADM2 results are found to be closer
to the 'REF', as compared to HTAP-MOZ results, suggesting that the RADM2 chemical mechanism is better
suited to simulate ozone over this region.

**5. Overall evaluation and recommendations**
In this section, we present a sub-regional evaluation of all simulations by subdividing the domain into five
geographical areas, i.e. North, South, East, West and central India, as shown in Fig. 1. The recommendations for
the individual stations based on the model evaluation are summarized in the Supplementary material (Table S1 and
S2). The temporal correlation coefficients of diurnally varying $O_3$, spatially averaged over each of the five
different sub-regions, are found to be reasonably high, generally exceeding 0.7 (Table 5). The r values for
individual sub-regions are found to be similar among the four simulations. For example, over north India the r
values vary from 0.86 to 0.90. The model performance differs among several sub-regions, with correlations being
lower for central India (r = 0.67-0.75). Since the latter is based on only one station associated with complex terrain
(Jabalpur), we suggest that observations over additional stations should be conducted to evaluate the model
performance in the central Indian region. As correlations are similar among different simulations, we focus on the
mean bias values especially around noontime (Table 6). Amongst the four different combinations of simulations
performed we find HTAP-RADM2 yields lowest noontime biases over north (MB = ~2.4 ppbv) and central India
(~0.9 ppbv). The S4RS-RADM2 combination is recommended for the east (MB ~15.3 ppbv) and South (MB ~6.5
ppbv) Indian regions. On the other hand, INTEX-RADM2 is found to yield better agreement with measurements
over western India (MB = ~8 ppbv). The recommendation for each region based solely on the ability to predict
noontime $O_3$ concentrations is summarized in table 7. These results show that the performance of emission
inventories is regionally different, and that these biases should be considered in utilizing model for assessment of
air quality and impacts on human health and crop yields.

We finally evaluate the different simulations in the context of the entire south Asian region. Figure 13 shows a
comparison of model results and measurements with diurnal box/whisker plots, combining all stations for the four
different simulations. As mentioned earlier, noontime ozone levels are overestimated by all four simulations. The



overestimation of noontime ozone is found to be largest in the HTAP-MOZ simulation, followed by HTAP-
RADM2, and lowest with S4RS-RADM2. These results further suggest that assessment of the tropospheric ozone
budget as well as implications for public health and crop loss are associated with considerable uncertainty, and
biases need to be considered. A recent study (Ghude et al., 2016), for example, subtracted 15 ppbv from the WRF-
Chem simulated ozone mixing ratios before deriving premature mortalities over the Indian region. The results of
this study are summarized in the form of a polar plot (Fig. 14) showing the monthly mean diurnal variation from
all runs for the entire south Asian domain. The noontime normalized mean bias values with respect to observed
values are ~11% (S4RS-RADM2), ~12.5% (INTEX-RADM2), ~22% (HTAP-RADM2) and ~36.5% (HTAP-
MOZ). It is interesting to note that the SEAC4RS inventory (representative of year 2012) yields quite similar
results as the INTEX-B inventory (representative of year 2006). It is concluded that the SEAC4RS inventory,
which is the most recent inventory amongst the three inventories considered in this study, is best suited for $O_3$
prediction over south Asian region as a whole in combination with RADM2 Chemistry.

**6. Summary and conclusions**
In this paper, we evaluated the WRF-Chem simulated surface ozone over South Asia during the pre-monsoon
season against recent in situ measurements from a network of 18 stations, employing three different inventories
(EDGAR-HTAP, INTEX-B, and SEAC4RS) for anthropogenic emissions with the RADM2 chemical mechanism.
WRF-Chem simulated ozone distributions showed highest ozone mixing ratios (~55 ppbv and higher) over
northern India and the Tibetan Plateau. In general, modelled average ozone distributions from different inventories
are found to be in agreement with previous studies over this region. Evaluation on diurnal time scales
demonstrates the ability of the model to reproduce observed $O_3$ patterns at urban and rural stations, showing strong
noontime ozone build-up and chemical titration and deposition loss during the night-time. WRF-Chem also
captures the smaller diurnal amplitudes observed over high altitude, relatively pristine stations. However, model
showed limitations in capturing ozone mixing ratios in the vicinity of the complex terrain, indicating that even a
relatively high horizontal resolution of 12 km x 12 km could not fully resolve the topography induced effects.
Overall WRF-Chem simulations show reasonable agreement with observations, with correlation coefficients
generally higher than 0.7 for most of the sites. It is found that the HTAP, INTEX-B and SEAC4RS inventories
lead to very similar distributions of 24-h average ozone over this region. However, noontime (1130-1630 IST) $O_3$
mixing ratios over continental South Asia differ significantly among the three inventories. HTAP inventory
generally leads to noontime $O_3$ mixing ratios higher by 10 ppbv over the Indo-Gangetic plain (IGP), 20 ppbv over
Central India, and 30 ppbv over Southern India, compared to the INTEX-B and SEAC4RS inventories. A
comparison of monthly average $O_3$ net production rate during 0630-1230 IST among the three inventories shows
that the HTAP emissions result in faster $O_3$ production (~9 ppbv h$^{-1}$) throughout the IGP region compared to the
other two inventories. Differences are also found over the southern Indian region with stronger ozone production
in HTAP, followed by INTEX-B and SEAC4RS. The results show similar 24-h average ozone distributions, but
large differences in noontime ozone build up, pointing to the uncertainties in emission inventories over this region.
We further investigated the sensitivity of modelled ozone to two extensively used chemical mechanisms, RADM2
and MOZART, and maintaining the HTAP emissions. Noontime average surface ozone distributions predicted by
MOZART show significant enhancements (10-15 ppbv) with respect to RADM2 over most of the Indian region,
except over western India. MOZART predicts higher ozone concentrations than RADM2 by up to 20 ppbv and



more over the South Indian region. Monthly average ozone mixing ratios are predicted to be higher by the
MOZART chemical mechanism compared to RADM2, as was also found over Europe (Mar et al., 2016). The
differences in ozone production between the MOZART and RADM2 chemical mechanisms are mainly attributed
to the additional chemical species and reactions, differences in the rate constants for several inorganic reactions,
and photolysis schemes used. A comparison of the monthly average chemical $O_3$ tendency (ppbv $h^{-1}$) during 0630-
1230 IST shows that in contrast with average $O_3$ mixing ratios, which were found to be higher in MOZART, the
net $O_3$ production rates at the surface are higher with RADM2 chemistry, especially over the IGP and central
India. The net $O_3$ production rates at the surface with RADM2 are found to be 6 to 9 ppbv $h^{-1}$, and higher over the
IGP, whereas these rates are generally lower with MOZART (4-8 ppbv $h^{-1}$), except in the northeastern IGP (>9
ppbv $h^{-1}$). Analysis of the vertical mixing tendency revealed that higher surface ozone mixing ratios in the
MOZART simulation are due to mixing with ozone rich air from aloft. Analysis of vertical distributions of
chemical $O_3$ tendencies reveals stronger photochemical production of ozone aloft with MOZART compared to
RADM2. Our analysis highlights the significant effects of the employed chemical mechanism on model predicted
ozone over South Asia.
Qualitatively, RADM2 and MOZART simulations predict similar diurnal patterns; however the absolute $O_3$
mixing ratios differ significantly. Noontime ozone mixing ratios predicted by MOZART are significantly higher at
12 out of 18 stations, while these were found to be similar at 6 stations. Over several urban and rural stations in
India MOZART is found to titrate ozone relatively strongly during the night, while producing higher or similar
ozone levels during noontime compared to RADM2. The contrasting evaluation results between day- (noon) and
night-time could counterbalance in evaluation studies limited to 24 h averages, possibly showing better agreement
and therefore hence it is pertinent to consider the diurnally resolved model performance. In general, the noontime
ozone mixing ratios predicted by RADM2 are found to be in better agreement with in situ measurements at the
surface compared to MOZART.
Model evaluation over different geographical regions in South Asia reveals strong spatial heterogeneity in the
WRF-Chem performance. SEAC4RS inventory leads to better agreement with observations over east (MB = ~15.3
ppbv) and south India (~6.5 ppbv), whereas the HTAP inventory performs better over north (MB = ~2.4 ppbv) and
central India (~0.9 ppbv), and INTEX-B over west India (MB = ~8 ppbv). For the entire region, the overestimation
of noontime ozone is found to be highest with the HTAP inventory (with the MOZART chemical mechanism) and
lowest with the SEAC4RS inventory. The noontime normalized mean bias is lowest for the SEAC4RS inventory
with the RADM2 chemical mechanism (~11%), followed by INTEX-B with RADM2 (~12.5%), HTAP with
RADM2 (~22%), and HTAP with MOZART (~36.5%). These results further suggest that the assessment of the
tropospheric ozone budget and consequently its implications on public health and agricultural output should be
carried out cautiously by considering the large uncertainties associated with use of emission inventories and
chemical mechanism incorporated. It is interesting to note that the SEAC4RS inventory (representative of 2012)
yields results comparable to the INTEX-B inventory (for 2006), even though the SECA4RS inventory has about
46% higher $NO_x$, 9% higher NMVOC and 15% lower CO emissions compared to INTEX-B. We conclude that the
SEAC4RS inventory, the most recent inventory amongst the three inventories, is best suited for $O_3$ prediction over
south Asian region as a whole in combination with RADM2 Chemistry. Our study highlights the need to also
evaluate $O_3$ precursors, similar to that conducted here for ozone, to further reduce uncertainties in modelled ozone
over South Asia for the better assessment of implications of surface ozone on public health and crop yield.





**Data availability**: The model output from all the numerical simulations is available at the MPG supercomputer HYDRA (http://www.mpcdf.mpg.de/services/computing/hydra) and would be provided by contacting the corresponding authors. The observed values shown for comparison are from previous papers with complete list of references provided in the Table 4. New observations for Delhi and Pune stations are available from the SAFAR program (http://safar.tropmet.res.in/).

**Acknowledgement**

A. Sharma acknowledges the fellowship from the Max Planck Institute for Chemistry to carry out this study. S. S. Gunthe acknowledges the support from DST-Max Planck partner group at IIT Madras and Ministry of Earth Sciences (MoES), Govt. of India. Model simulations have been performed on the MPG supercomputer HYDRA (http://www.mpcdf.mpg.de/services/ computing/hydra). Initial and boundary conditions data for meteorological fields were obtained from ECMWF website (http://www.ecmwf.int/en/research/climate-reanalysis/era-interim). The HTAP v2 anthropogenic emissions were obtained from http://edgar.jrc.ec.europa.eu/htap_v2/ index.php?SECURE=123. Authors are grateful to Yafang Cheng (MPI-C) for providing SEAC4RS emission. The INTEX-B anthropogenic emissions were obtained from http://bio.cgrer.uiowa.edu/EMISSION_DATA_new /data/intex-b_emissions/. MOZART-4/ GEOS5 output used as initial and boundary conditions for chemical fields is acknowledged. The pre-processors and inputs for biogenic and biomass-burning emissions were obtained from NCAR Atmospheric Chemistry website (http://www.acd.ucar.edu/wrf-chem/). Authors are also thankful for the usage of HPC supercluster and to the staff at P. G. Senapathy Computer Center at IIT Madras.

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





**Table 1.** Abbreviations/Acronym

| | |
|---|---|
| EDGAR | Emission Database for Global Atmospheric Research |
| HTAP | Hemispheric Transport of Air Pollution |
| IGP | Indo Gangetic plain |
| IST | Indian standard time |
| INTEX-B | Intercontinental Chemical Transport Experiment Phase B |
| MB | Mean Bias |
| MOZART | Model for Ozone and Related Chemical Tracers |
| NMB | Normalized mean bias |
| PBL | Planetary boundary layer |
| RMSD | Centered root mean squared difference |
| RRTM | Rapid Radiative Transfer Model |
| SEAC4RS | Southeast Asia Composition, Cloud, Climate Coupling Regional Study |
| WRF-Chem | Weather research and forecasting model coupled with chemistry |

**Table 2.** Sub-regional estimates of anthropogenic emissions (in million mol h$^{-1}$) in the three emission inventories used

| Region | HTAP | | | INTEX-B | | | SEAC4RS | | |
|---|---|---|---|---|---|---|---|---|---|
| | $NO_x$ | NMVOC | CO | $NO_x$ | NMVOC | CO | $NO_x$ | NMVOC | CO |
| North | 8.1 | 14.0 | 110.0 | 6.3 | 10.0 | 96.1 | 8.7 | 10.7 | 86.9 |
| East | 5.8 | 10.1 | 102.9 | 6.0 | 6.9 | 78.8 | 6.7 | 8.2 | 72.4 |
| West | 2.9 | 4.6 | 31.0 | 1.8 | 2.1 | 24.7 | 3.7 | 2.9 | 24.3 |
| Central | 4.6 | 4.2 | 44.6 | 2.0 | 2.9 | 34.7 | 4.9 | 3.1 | 26.2 |
| South | 5.4 | 5.8 | 37.2 | 2.7 | 4.1 | 46.2 | 3.5 | 3.4 | 28.3 |
| Total | 26.8 | 38.7 | 325.7 | 18.8 | 26.0 | 280.5 | 27.5 | 28.3 | 238 |

**Table 3.** A brief description of the different WRF-Chem simulations conducted

| Sr. No. | Simulation name | Emission Inventory | Year of Emission Inventory | Spatial Resolution of Emission Inventory | Chemical Mechanism |
|---|---|---|---|---|---|
| 1 | HTAP-RADM2 | HTAP | 2010 | 0.1$^o$x 0.1$^o$ | RADM2 |
| 2 | INTEX-RADM2 | INTEX-B | 2006 | 0.5$^o$x 0.5$^o$ | RADM2 |
| 3 | S4RS-RADM2 | SEAC4RS | 2012 | 0.1$^o$x 0.1$^o$ | RADM2 |
| 4 | HTAP-MOZ | HTAP | 2010 | 0.1$^o$x 0.1$^o$ | MOZART-4 |





**Table 4.** List of observation sites and data sources used. Site nomenclature in brackets in column 1 is used in figures 1, 5, 6, 9 and 10.

| Site | Type | Latitude | Longitude | Altitude (m.a.s.l) | Data period | Reference |
|------|------|----------|-----------|--------------------|-------------|-----------|
| Mohali (MOH) | Urban | 30.7°N | 76.7°N | 310 | May 2012 | Sinha et al. (2014) |
| Nainital (NTL) | Highly complex | 29.37°N | 79.45°E | 1958 | Apr 2011 | Sarangi et. al. (2014) |
| Pantnagar (PNT) | Urban/ complex | 29.0°N | 79.5°E | 231 | Apr 2009-11 | Ojha et al. (2012) |
| Delhi (DEL) | Urban | 28.65°N | 77.27°E | 220 | Apr 2013 | SAFAR data |
| Dibrugarh (DBG) | Rural/ complex | 27.4°N | 94.9°E | 111 | Apr 2010-13 | Bhuyan et al. (2014) |
| Darjeeling* | Complex | 27.01°N | 88.25°E | 2134 | Apr 2004 | Lal (2007) |
| Kanpur (KNP) | Urban | 26.46°N | 80.33°E | 125 | Mar-May 2010-13 | Gaur et al. (2014) |
| Mt. Abu (ABU) | Highly complex | 24.6°N | 72.7°E | 1680 | Apr 1993-2000 | Naja et al. (2003) |
| Udaipur (UDP) | Urban | 24.58°N | 73.68°E | 598 | Apr 2010 | Yadav et al. (2014) |
| Jabalpur (JBL) | Complex | 23.17°N | 79.92°E | 411 | Apr 2013 | Sarkar et al. (2015) |
| Ahmedabad (ABD) | Urban | 23.03°N | 72.58°E | 53 | May 2011 | Mallik et al. (2015) |
| Haldia (HAL) | Urban/ coastal | 22.05°N | 88.03°E | 8 | Apr 2004 | Purkait et al. (2009) |
| Bhubaneshwar (BBR) | Urban | 21.25°N | 85.25°E | 45 | Mar-May 2010 | Mahapatra et al. (2012) |
| Joharapur (JHP) | Rural | 19.3°N | 75.2°E | 474 | Apr 2002-2004 | Debaje et al. (2006) |
| Pune (PUN) | Urban | 18.54°N | 73.81°E | 559 | Mar-May 2013 | SAFAR data |
| Anantapur (ANP) | Rural | 14.62°N | 77.65°E | 331 | Apr 2009 | Reddy et al. (2010) |
| Gadanki (GDK) | Rural | 13.48°N | 79.18°E | 375 | Mar-May 2010-11 | Renuka et al. (2014) |
| Kannur (KNR) | Rural/ coastal | 11.9°N | 75.4°E | 5 | Apr 2010 | Nishanth et al. (2012) |
| Thumba/ Trivendrum (TRI) | Urban/ coastal | 8.55°N | 77°E | 3 | Apr 2009 | David et al. (2011) |

\* At Darjeeling only monthly mean value is available.

**Table 5.** A comparison of correlation coefficients (r) over different regions for the four simulations

| Region | HTAP-RADM2 | INTEX-RADM2 | S4RS-RADM2 | HTAP-MOZ |
|--------|-----------|-------------|------------|----------|
| North | 0.90 | 0.86 | 0.88 | 0.90 |
| East | 0.98 | 0.97 | 0.97 | 0.98 |
| West | 0.99 | 0.98 | 0.98 | 0.99 |
| Central | 0.70 | 0.67 | 0.69 | 0.75 |
| South | 0.99 | 0.98 | 0.97 | 0.97 |
| Overall | 0.98 | 0.97 | 0.97 | 0.99 |

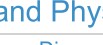
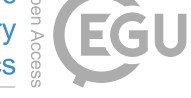

768 **Table 6.** A comparison of noontime (1130-1630 IST) average mean biases in ppbv over different regions for the
769 four simulations

| Region | HTAP-RADM2 | INTEX-RADM2 | S4RS-RADM2 | HTAP-MOZ |
|--------|-----------|-------------|-----------|----------|
| North | 2.4 | -3.3 | -4.1 | 8.3 |
| East | 19.5 | 19.5 | 15.3 | 29.9 |
| West | 11.4 | 8.0 | 9.0 | 14.0 |
| Central | 0.9 | -8.0 | -2.5 | 8.8 |
| South | 15.3 | 8.2 | 6.5 | 25.5 |
| Overall | 10.5 | 5.9 | 5.2 | 17.3 |

774 **Table 7.** Recommendations based on noontime average mean biases over different regions for the four
775 simulations

| Region | HTAP-RADM2 | INTEX-RADM2 | S4RS-RAMD2 | HTAP-MOZ |
|--------|-----------|-------------|-----------|----------|
| North | √ | | | |
| East | | | √ | |
| West | | √ | | |
| Central | √ | | | |
| South | | | √ | |
| Overall | | | √ | |

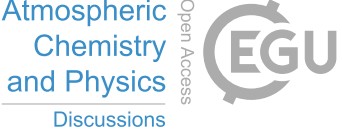



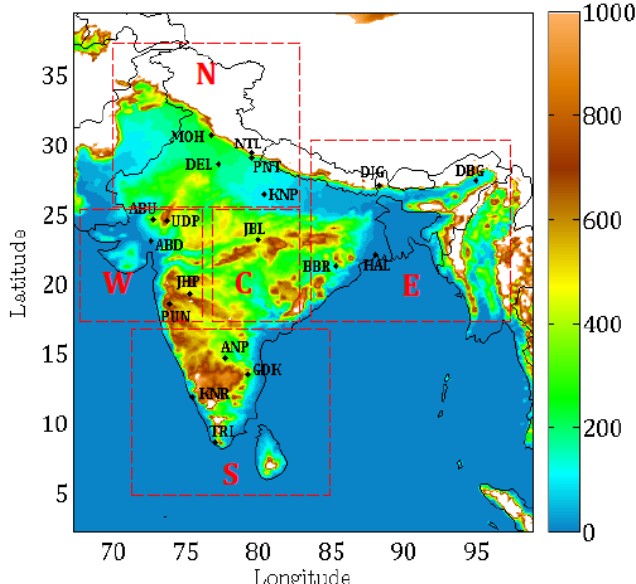

**Figure 1**. Simulation domain showing terrain height (in metres) and observation sites. White region indicates that the terrain height is equal to or exceeds 1 km. The domain is subdivided into five regions viz. North (N), South (S), East (E), West (W) and central (C) regions, as shown by red rectangles.



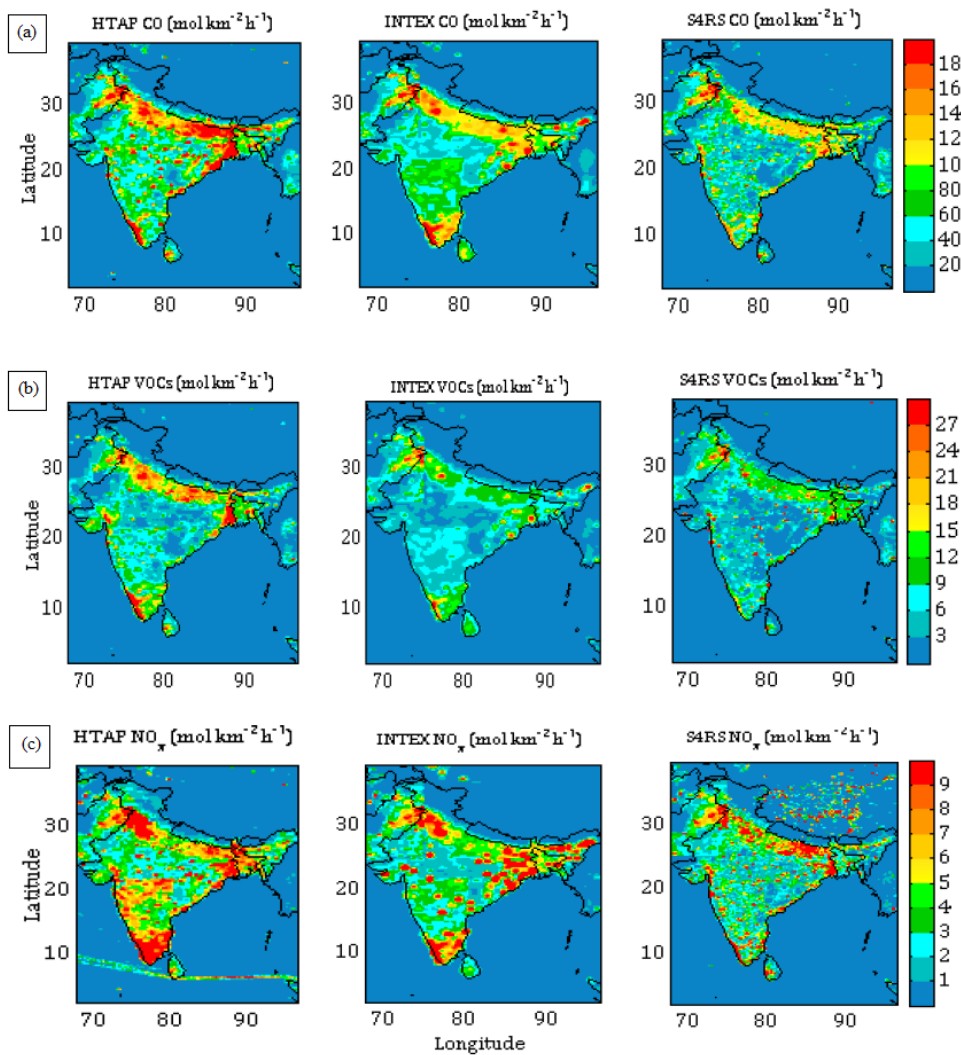


**Figure 2.** A comparison of (a) CO, (b) NM VOC and (c) NO$_x$ emissions between the three inventories used (see Section-2.2 for description).













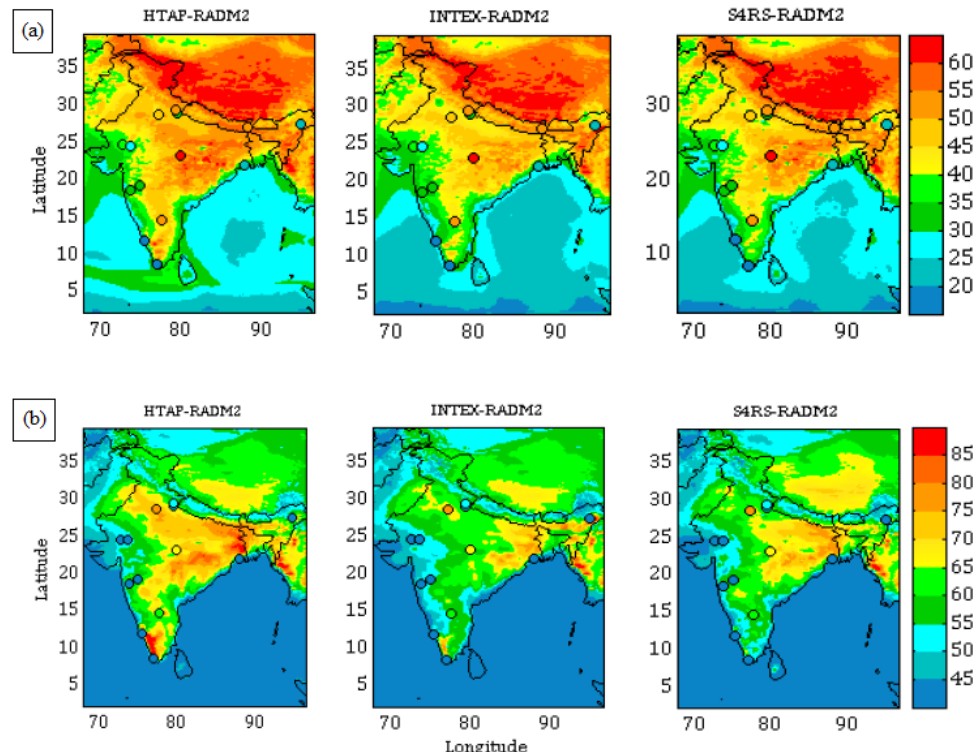

**Figure 3.** Spatial distribution of monthly (for April) average surface ozone calculated for (a) 24 h and (b) noontime (1130-1630
IST). The average ozone mixing ratios (ppbv) from observations are also shown for comparison on the same colour scale. Note
the difference in colour scales in the top and bottom rows.





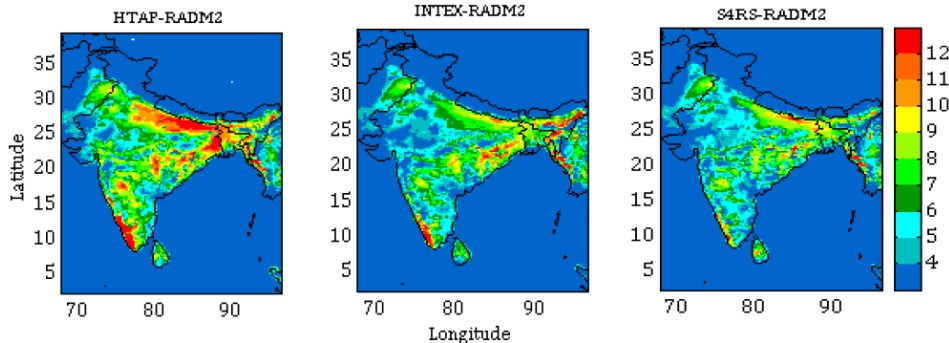

**Figure 4.** Spatial distribution of net daytime surface ozone chemical tendency (in ppbv h⁻¹) for the month April during 0630-
1230 IST





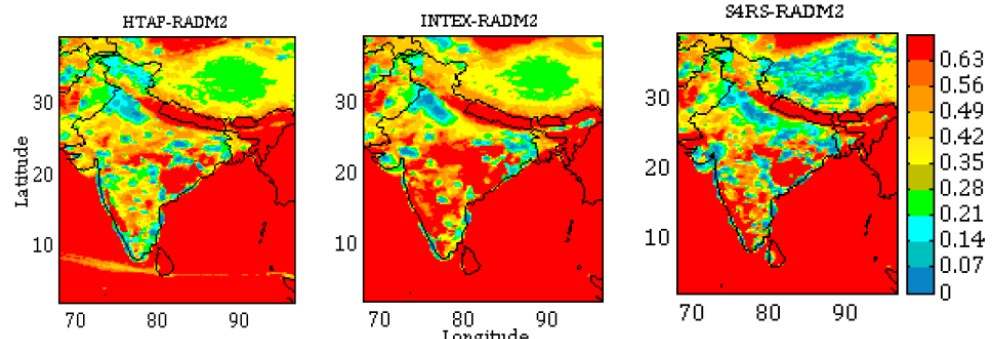


**Figure 5.** Spatial distribution of net daytime surface CH$_2$O to NO$_y$ ratio in simulations with different inventories for the month
April during 0630-1230 IST




**Figure 6.** Comparison of monthly average diurnal variation of surface ozone simulated using different emission inventories at various observation sites. The observational data is available for the period indicated in the figure whereas all model simulations are for the year 2013. Error bars represent the temporal standard deviations of the monthly averages. All model simulations are with RADM2 chemistry.




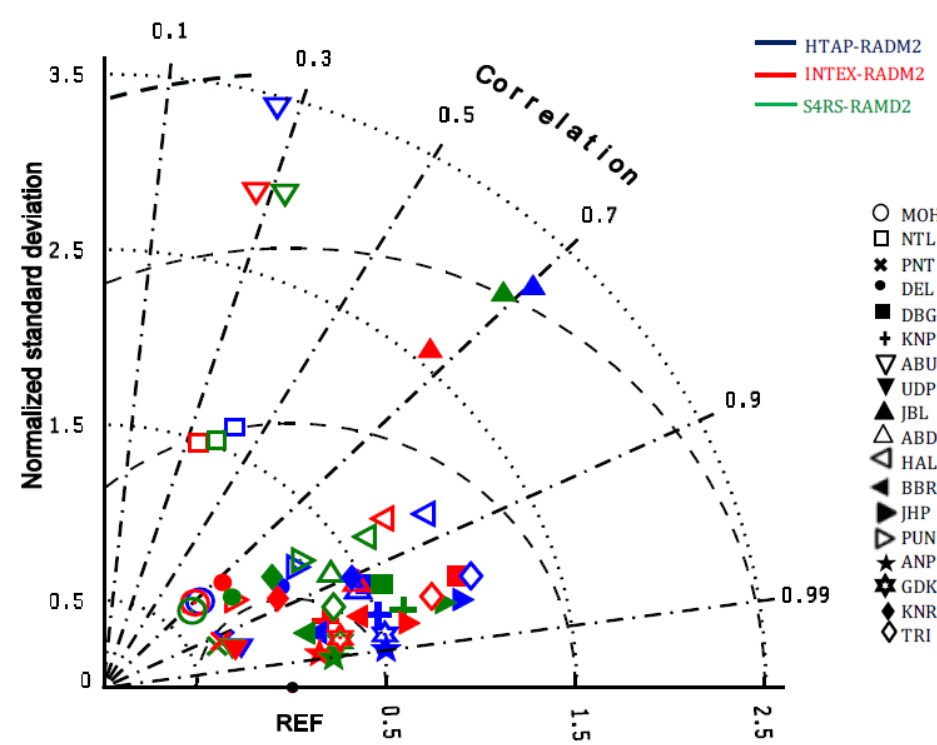


**Figure 7.** Taylor diagram showing model statistics (r, normalized standard deviation and RMSD) at all sites. The correlation is the cosine of the angle from the horizontal axis, the root mean square difference is the distance from the reference point (REF) and the standard deviation is the distance from the origin.













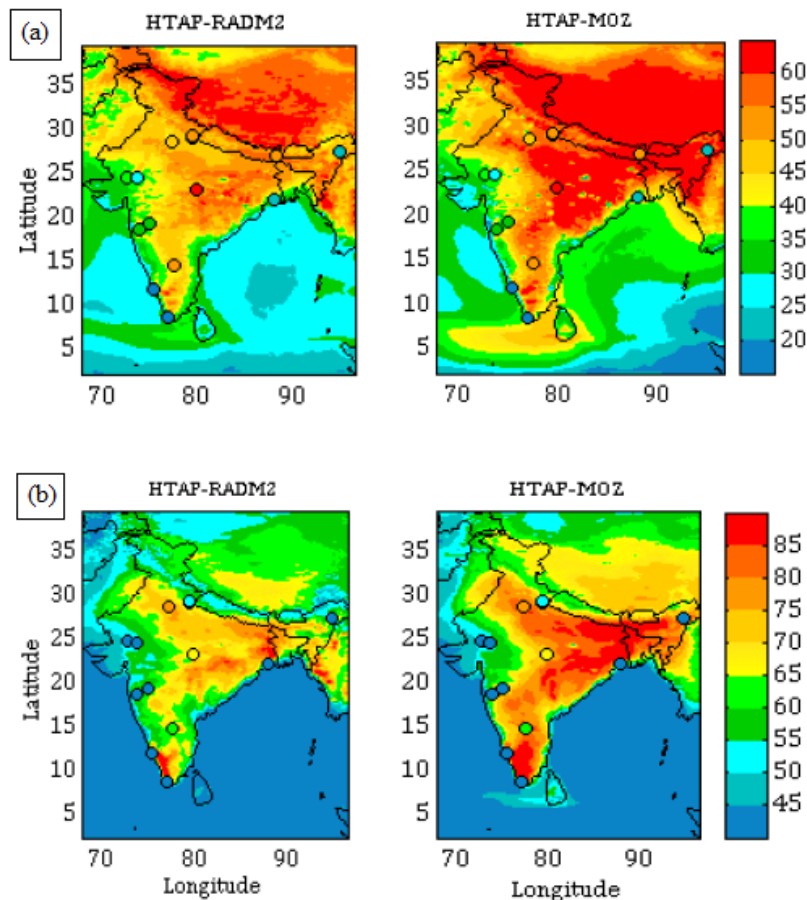

**Figure 8.** Spatial distribution of monthly (April) average surface ozone calculated for (a) 24 h and (b) noontime (1130-1630 IST), comparing the chemical mechanisms (RADM2 and MOZART). The average ozone mixing ratios (ppbv) from observations are also shown for comparison on the same colour scale. Note the difference in colour scales in the top and bottom rows.






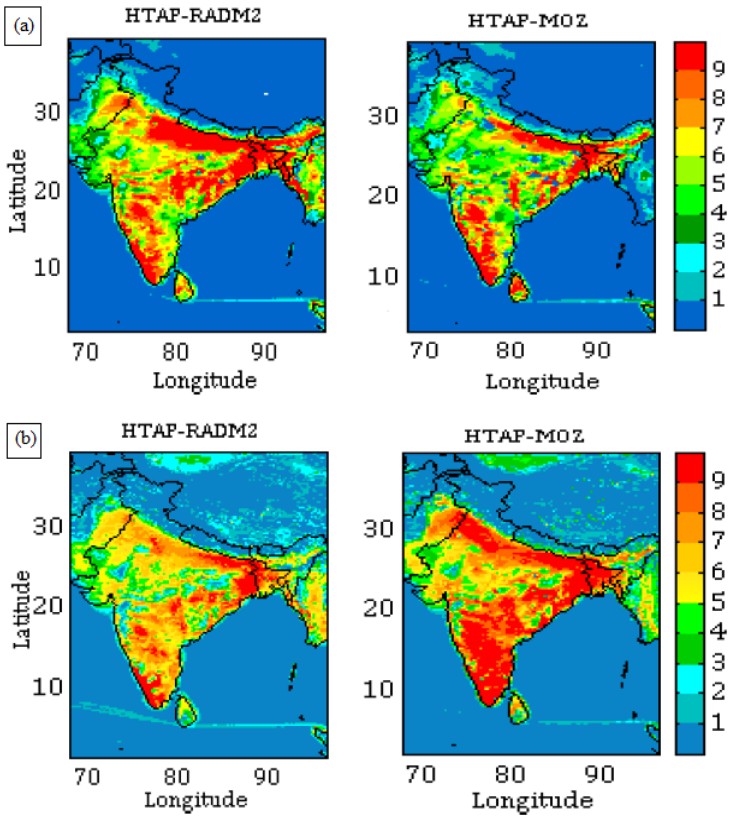

**Figure 9.**  Spatial distribution of average **(a)** net daytime surface ozone chemical tendency (in ppbv h$^{-1}$) **(b)** net daytime surface
ozone chemical +vertical mixing tendency (in ppbv h$^{-1}$) for April during 0630-1230 IST





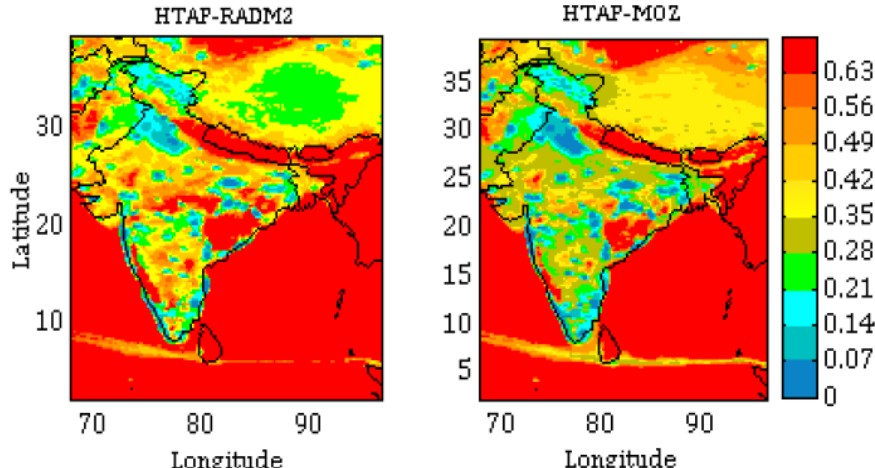

**Figure 10.** Spatial distribution of net daytime surface $CH_2O$ to $NO_y$ ratio in simulations with different chemical mechanisms
for the month April during 0630-1230 IST




**Figure 11.** Comparison of monthly average diurnal variation of surface ozone simulated using different chemical mechanisms
at various observation sites. The observational data is available for the period indicated in the figure whereas all the model
simulations are for the year 2013. Error bars represent the temporal standard deviations of the monthly averages. All model
simulations are with HTAP inventory.



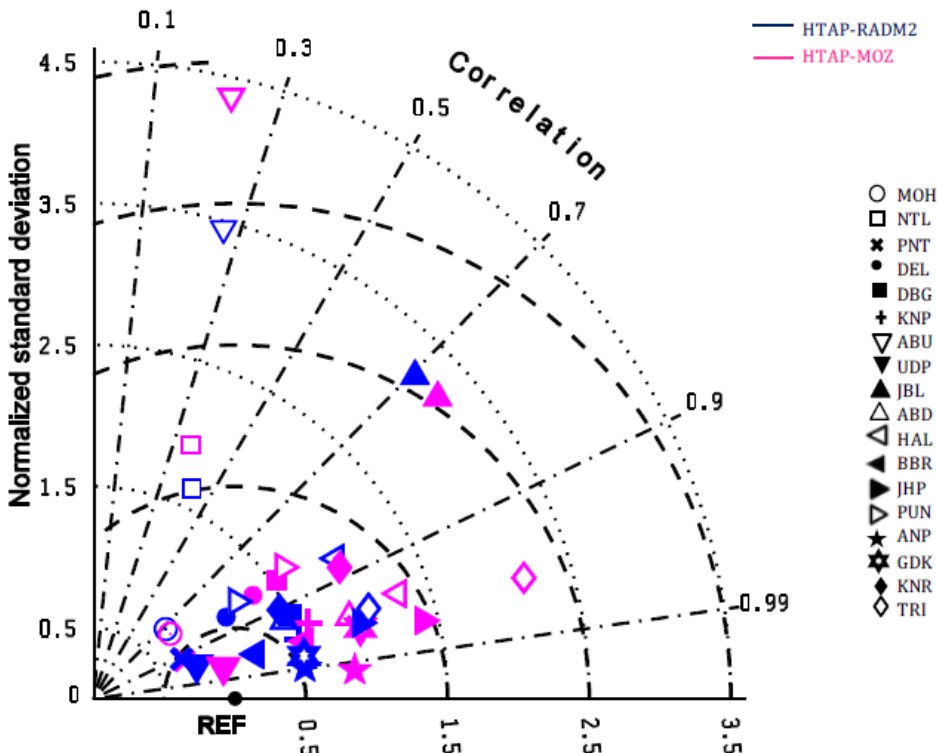


**Figure 12.** Taylor diagram showing model statistics (r, normalized standard deviation and RMSD) at all sites. The correlation is
the cosine of the angle from the horizontal axis, the root mean square difference is the distance from the reference point (REF)
and the standard deviation is the distance from the origin.







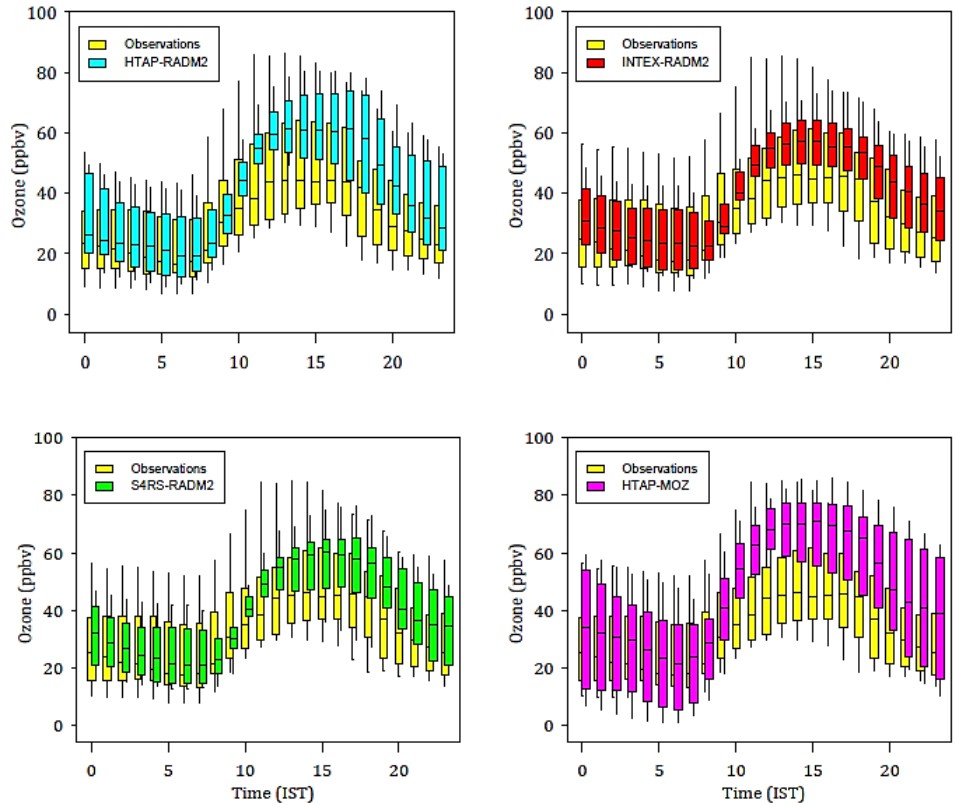


**Figure 13.** A box/whisker plot comparison of monthly average diurnal variation of surface ozone from model runs and observations over the entire domain (after spatially averaging the results). Upper and lower boundaries of boxes denote the 75th and 25th percentiles and whiskers represent the 95th and 5th percentiles. The line inside the box is the median.






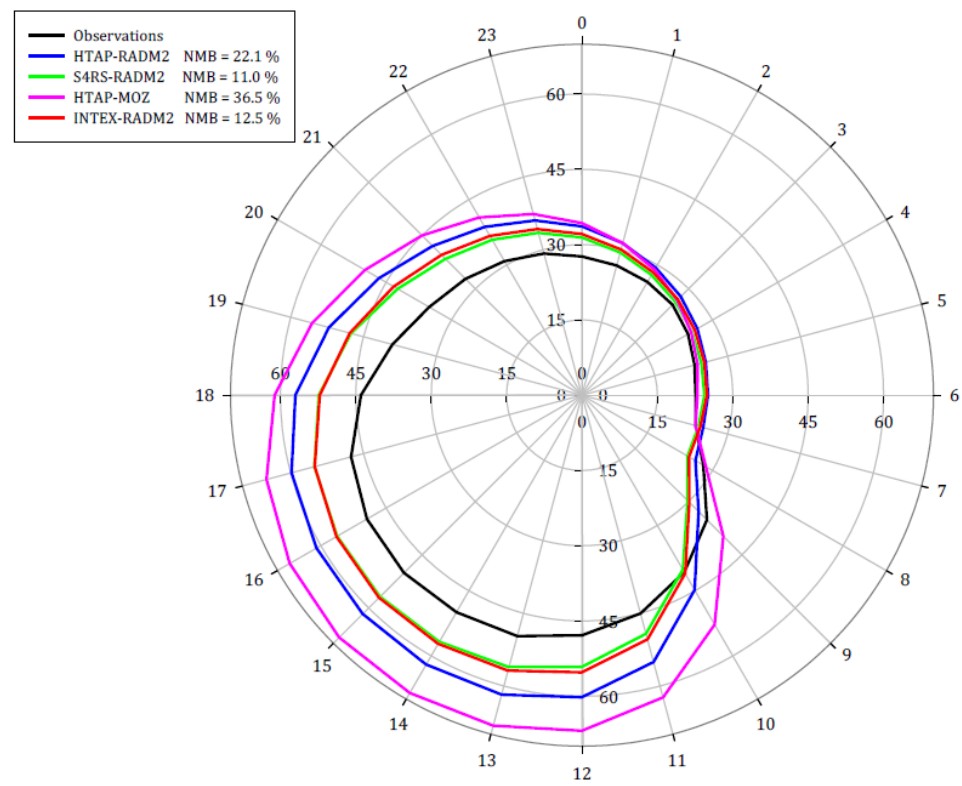

**Figure 14.** Polar plot for monthly mean diurnal variation of surface ozone (in ppbv) from all model simulations and
observations each spatially averaged over all sites. The numbers on the outermost circle represent the hour of the day and the
radial distance from the centre represents surface ozone mixing ratios in ppbv. The normalized mean bias (NMB in %) values
in all the simulations are also provided in the caption box.