# Peer review of "WRF-Chem simulated surface ozone over South Asia during"

_Atmospheric Chemistry and Physics, 2016_

## Referee Comment (RC1) · Anonymous Referee #1 · 12 Jan 2017

The paper describes uncertainty of modeled ozone to emission inventories of precursors generated by three different international effort. An evaluation of two chemical mechanisms MOZART and RADM-2 are also presented for one of the inventories. Results for April 2013 are presented. As presented it is a fairly unconstrained problem in terms of evaluation of the goodness of one emission field over the other purely based on ozone alone. I have tried to learn something new from the manuscript that I could have not guessed by just looking at table 1. They all have about the same NOX and HTAP has nearly 50% more NMVOC's than the other two emissions. If we are in a hydrocarbon limited regions (as it seems like most of India is) then HTAP will produce more ozone. I don't see the mystery in this conclusion. Fixing emissions to get the cor-

rect answer is patently wrong in a situation like here, where there so many physical and chemical process unknowns. (a) It would have been very useful if we could have some figures showing comparison between observed and measured hydrocarbon. I am sure, we will get the answer that there are not any. I would suggest that the group should collect some data on NMHC's to support this analysis if that were the case. (b) Where is the evaluation of NOX simulated at these sites? I have never seen a ozone evaluation paper that completely ignores the precursor observations and entirely based on ozone measurement. The comparison between MOZART and RADM-2 also hinges on an unknown in the model performance over India. I have seem a few papers on WRF from India that shows huge ( +/- 1000 mts or more) differences in PBL heights by just using two different PBL schemes in the model. If MOZART is producing more ozone in the upper troposphere and is getting entrained into the PBL, where is the evaluation of PBL heights or entrainment rates in the study. (a) Why is MOZART producing more ozone in the upper troposphere than RADM? Is it because the photolysis rates used in RADM different than the ones used in MOZART? (b) I am guessing the photolysis code used for both RADM and MOZART are the same – but please check. (c) It seems like the ensemble based cloud scheme (GD) doesn't perform well over India. It has too much downward flux of air from the upper troposphere to surface. I recommend you try with a different scheme or carefully evaluate the UT/PBL fluxes in the model with observations.

(d) I have also noticed that lines 130/131 probably refer to spectral nudging and not really a FDDA. Do you have or assimilated any observational meteorological data from the Indian Meteorological Department (sondes, surface weather stations etc) to perform the FDDA?

(e) Performing spectral nudging to ERA probably is not a good idea, unless you can establish that it is a good representation of synoptic scale conditions over India during this period. Many instances (specially at 12 km resolution) it is better to run the model in data poor areas with model physics than nudging the entire wind profile to ERA or

any other reanalysis. Have you evaluated the model synoptic scale meteorology for the simulation period with any observations?

(f) Line 85/86 cites a paper that show the differences between simulated ozone is 4.5% with different emissions. Is the goal to improve upon that. I personally will be quite happy if you can predict ozone at less than 5% accuracy using a model.

(g) A Taylor diagram makes lots of sense when you are trying to find out which model (or model physics) is getting close to a reference point. Emissions by themselves have no real value and improving them is not really a model issue, more of an inventory developers problem. I don't see the point of this as the errors could be due to any number of physics or chemistry issues and not related to emissions at all. I can simply scale the HTAP emissions to a lower value and get closer to the other two emissions, that doesn't lead to a model improvement.

(h) The metric CH2O/NOy was presented in several figures. What am I supposed to learn from this? I am guessing the RADM scheme has no methane and MOZART has methane in its chemical trace list. How is NOy defined, does it include HNO3? The variability you see is most likely because of different loading of NMHC from each emission. Doesn't tell much about anything in my opinion.

(i) During this time of the year the atmosphere over the central plains in India is loaded with dust. What role does dust play in the ozone production / removal?

(j) The biomass burning identified has a major source of precursors also produces copious amounts of aerosols and in particular brown carbon. Brown carbon can change photolysis rates quite significantly and reduce ozone formation. How much of the disagreement is due to not accounting for these types of effects that are unique to India? We may have to fix these issues before trying to fix emissions. This only adds one more bad scientific processes to an already poor decision making in India for pollution control.

(k) Have you evaluated the water vapor in the model during these months. Does the error in water vapor in the model explain some of the differences?

---

## Referee Comment (RC2) · Anonymous Referee #2 · 5 Mar 2017

**A review report on the ACPD manuscript entitled "WRF-Chem simulated surface ozone over South Asia during the pre-monsoon: Effects of emission inventories and chemical mechanisms" by Sharma et al., 2016.**

**General**

The study investigates simulated ozone over South Asia, using several simulation scenarios, composed of different inventories and chemical mechanisms. The simulation results were evaluated using data from an in-situ monitoring network. Among the findings of the study is that simulated daytime ozone maximum differ significantly between different emission scenarios, by as high as -22%, in contrast to the 24h mean values, which are more consistent. The results are not surprising, especially on local scale, given that measured ozone is primarily photo-chemically formed. However, a major issue here is that the authors use different temporal emissions (2010 for HTAP, 2006 for INTEX-B) form different emission inventories and are trying to validate the model simulations of 2013 (using reanalysis ECMWF product) with measurements from completely different temporal period (e.g, 2004 or before, and 2009-2013), except for 4 stations. The authors should clarify the significances of these results in this context, especially in this very active developing region? Impacts from biomass-burning emissions are not adequately discussed. The authors proclaim similar results between different emissions scenarios despite the different temporal periods. However, these claimed similarities should be only a warning of some compensating effects that cancel the interesting differences caused by the emissions annual trends and variability.

The study sounds scientifically interesting and well written, but still need more consistent analysis and casual discussions on the driving factors of the differences between these scenarios.

**Specific comments**

Page 1, lines 32-33. The conclusion that the SEAC4RS-RADM2 scenario preforms better than the others does not sound novel scientific information. I think that it is important here that the authors shed some light on why this specific scenario works better than the others.

Page 3, lines 103-: The authors mentioned high pollution loading and biomass burning as reasons for the intense ozone photochemical formation during the pre-monsoon period. It would be also very interesting if the authors could investigate how biomass burning emissions and transport affect ozone photochemical formation in the study's domain.

Page 4, lines 139-141: Could the authors elaborate on the difference between the two aerosol modules used, the (MADE/ SORGAM) vs GOCART, and how this would affect their results?

Page 4, lines 142-145: Also, how the different photolysis schemes Fast-J and F-TUV may affect the results?

Could the authors employ the same aerosol and photolysis scheme for each scenario (using different emissions and chemical mechanism), so that casual factors for the differences can be determined?

Page 4, line 152: What is the effect of using year 2010 HTAP emissions as opposed to experimental observation date and model reanalysis of 2013? How this may affect their conclusions?

Page 5, line 160: What is the effect of using year 2006 INTEX-B emissions as opposed to experimental observation date and model reanalysis of 2013? How the authors account for using emissions from different years?, especially in this high-pace developing region?

Page 6, lines 198-200: But how the comparison would make sense given that the emissions are from different years and are also different between different inventories?

Page 6, line 204: No, that too much difference, I do not think the authors can use (2004 or before) ozone measurements to validate model simulations for years 2013 using emissions from different temporal periods?? I think the authors need to reconsider all these comparisons..

Page 6, lines 219-220: Could the authors provide quantitative numbers for this similarity between HTAP, INTEX and S4RS scenarios (e.g., r^2)? To me, they look quantitatively different..

Page 7, lines 241-250: Again, it is important to address here if the differences in the ozone production rates between different emission scenarios are related to using different temporal periods for the emission inventories or related to different emission inventories as it appears here?

Page 8, lines 304-318: So, are these differences related to chemical mechanism, or the constrained different overhead ozone column, or photolysis rates (Fast-J vs F-TUV) or different aerosol modules (static vs dynmic)?

Page 11, lines 403-406: The authors claim interesting similar results despite the use of different temporal emission, but I think that shows only possible compensating effects that lead to the claimed similar results despite different emissions… I think that the authors should seriously address this issue as it significantly affect the credibility of the results.

Page 11, 420: Again, I still not convinced by the "overall agreement", given that the model is constrained to emissions from different temporal periods than the measurements as well as the model simulations (using reanalysis products from year 2013).

---

## Author Comment (AC1) · 31 May 2017

The response to the reviewer's comments, and the revised manuscript are uploaded as the Supplement zip file.

Please also note the supplement to this comment:
http://www.atmos-chem-phys-discuss.net/acp-2016-1083/acp-2016-1083-AC1-supplement.zip

---

## Author Response (AR1)

Response to referee comments on "WRF-Chem simulated surface ozone over South Asia during the pre-monsoon: Effects of emission inventories and chemical mechanisms" by A. Sharma et al.

**Anonymous Referee #1**

**Comment 1:** The paper describes uncertainty of modeled ozone to emission inventories of precursors generated by three different international effort. An evaluation of two chemical mechanisms MOZART and RADM-2 are also presented for one of the inventories. Results for April 2013 are presented. As presented it is a fairly unconstrained problem in terms of evaluation of the goodness of one emission field over the other purely based on ozone alone. I have tried to learn something new from the manuscript that I could have not guessed by just looking at table 1. They all have about the same NOX and HTAP has nearly 50% more NMVOC's than the other two emissions.

Response 1: We believe that the referee made the comparison between total emissions aggregated over all regions *in the table 2* (as Table 1 is showing abbreviations/acronyms). HTAP has about 43% and SEAC4RS has about 46% higher NOx as compared to the INTEX-B inventory. Hence the NOx emissions are not quite the same. Additionally SEAC4RS, the newest inventory of the three, has similar NOx levels to HTAP whereas it has similar VOC emissions as INTEX-B (the oldest inventory of the three). Considering the non-linear dependence of  $O_3$  formation on precursors, a set of numerical experiments is necessary to assess the influence of such large differences among the inventories. This information is added in the revised manuscript (Page:5, Lines:178-184). Finally, we explicitly emphasize the region-based evaluations of simulated ozone, and the differences in NOx emissions over regions are as high as 200% (South – INTEX-B vs. HTAP; Central – INTEX-B vs. SEAC4RS, etc.).

**Comment 2:** If we are in a hydrocarbon limited regions (as it seems like most of India is) then HTAP will produce more ozone. I don't see the mystery in this conclusion. Fixing emissions to get the correct answer is patently wrong in a situation like here, where there so many physical and chemical process unknowns.

Response 2: Here, reviewer is mentioning ozone formation over the Indian region as hydrocarbon-limited, which is quite contrary to what we have reported. This highlights again the importance of studies presenting numerical experiments as compared to concluding ozone production simply by comparing emission values.

Ozone production over most of the Indian region is NOx limited in INTEX-RADM2 simulation, as shown using the CH2O/NOy ratio (Figure 5). This result is in agreement with a previous study using this inventory (Kumar et al 2012b). In contrast, ozone production is relatively more sensitive to VOCs in the HTAP-RADM2 and S4RS-RADM2 simulations, with significant parts of the Indian region still being NOx limited. We suggest that our evaluation results should therefore be considered while analysing the surface ozone pollution, budget and impacts with any of the inventories or chemical mechanisms utilised in our paper over India.

We do not agree with the reviewer that many physical and chemical processes are unconstrained/unknown here. It is to be noted that the WRF-Chem model has been extensively used to successfully reproduce the meteorology and dynamics over this region. This is discussed with numerous references in the introduction section of our paper already (Page: 2-3, Lines: 69-83). For example, Kumar et al. (2012a) explicitly conclude that the meteorology is of sufficient quality to simulate the ozone chemistry over South Asia. It is to be noted that our configuration of the model setup is based on the findings of previous studies. In addition, nudging with ERA interim reanalysis here provides constraints to the simulated meteorology/dynamics.

The suggestion of the reviewer to evaluate additional schemes for boundary layer dynamics and convection has been incorporated in the revised manuscript (see response to your comments 4 and 6).

**Comment 3:** It would have been very useful if we could have some figures showing comparison between observed and measured hydrocarbon. I am sure, we will get the answer that there are not any. I would suggest that the group should collect some data on NMHC's to support this analysis if that were the case. (b) Where is the evaluation of NOX simulated at these sites? I have never seen a ozone evaluation paper that completely ignores the precursor observations and entirely based on ozone measurement.

Response 3: We agree that there is a need to conduct the measurements for precursors over this region. However this is beyond the objectives and the possibilities of the present study as described in the manuscript (Page: 3; Lines: 93—99). The evaluation of precursors would certainly provide further information about the uncertainties in the inventories and should be a recommended next step (Page:1, Line: 33-34; Page:14, Lines: 543—545), however, our conclusions assessing the simulated ozone would not be affected, which are given as follows:

(a) noontime ozone in the model significantly differs among different inventories (and also different chemical mechanisms) in contrast with the 24-h mean values, and that the current estimates of ozone impacts on human health and crop yield over South Asia have large uncertainties.

b) Ozone simulated using the SEAC4RS inventory (latest) coupled with RADM2 chemistry is in better agreement with observations making it more suitable for simulating surface ozone relative to other inventories used in the study.

We agree that there are very limited observations of precursors, nevertheless following reviewer's suggestion, we include an evaluation of modelled NOx, ethane and ethene against recent measurements (Table C1; Table S1 in revised Supplement). Significant differences are seen in NOx mixing ratios at Delhi, with only INTEX-RADM2 being within 1 standard deviation of the observed value. Ozone production at Delhi is VOC limited in all simulations in the present study (seen from CH2O/NOy ratio in Fig. 5). This indicates the importance of conducting measurements of NMVOCs in the Delhi region. At Kanpur also NOx from INTEX-RADM2 compares better with the observed values. At Mt. Abu in the west, NOx from HTAP-RADM2 compares better with observed values, however it should be noted that the site is also impacted by transported ozone during spring (Naja et al., 2003). At Udaipur, all simulations tend to underpredict NOx. At Haldia in the east, NOx from S4RS-RADM2 compares better with observed value and is seen to be within 1 standard deviation variability of the observed value in all simulations.

Modelled ethane mixing ratios are quite similar in all simulations and agree well with observed values at Mt. Abu but are underpredicted at Nainital by a factor of about 2. On the other hand,

modelled ethene mixing ratios at both Mt. Abu and Nainital agree relatively well with observed values in INTEX-RADM2 and S4RS–RADM2 as compared to HTAP-RADM2. The corresponding table and a small description is now added in the revised manuscript (Page: 6-7; Lines: 235-239 in the manuscript and Table S1; Section S1 on Page: 1-2 in revised supplement).

We would again like to mention that the observations of precursors are very sparse in the south Asian region and it is important to have an evaluation over a network of observations, as we present for ozone in this study, to understand their contribution into ozone formation and also the budget of NMVOCs over the region. However this does not affect the conclusions of the present study.

| Specie   | Site     | Reference                                  | Observations | HTAP- | INTEX- | S4RS- | HTAP- |
|----------|----------|--------------------------------------------|--------------|-------|--------|-------|-------|
|          |          |                                            | ±1 σ std     | RADM2 | RADM2  | RADM2 | MOZ   |
|          | Delhi    | SAFAR data                                 | 59.8±27.5    | 208.7 | 64.4   | 187.2 | 188.9 |
|          | Kanpur   | Gaur et al. (2014)                         | 5.0          | 10.2  | 6.5    | 30.5  | 9.1   |
| NOx      | Mt. Abu  | Naja et al. (2003)/
Kumar et al (2012b) | 2.1          | 1.7   | 1.1    | 1.1   | 1.4   |
|          | Udaipur  | Yadav et al. (2014)                        | 8.7±4.2      | 2.1   | 1.6    | 1.5   | 2.0   |
|          | Haldia   | Purkait et al. (2008)                      | 12.6         | 4.4   | 3.5    | 8.2   | 4.6   |
| NOy      | Nainital | Sarangi et al. (2014)                      | 1.8±1.6      | 3.2   | 2.7    | 2.9   | 2.6   |
| NMVOC    | Nainital | Sarangi et al. (2016)                      | 2.3          | 1.2   | 1.2    | 1.1   | 1.0   |
| (ethane) | Mt. Abu  | Sahu and Lal (2006)                        | 1.3          | 1.1   | 1.1    | 1.1   | 1.0   |
| NMVOC    | Nainital | Sarangi et al. (2016)                      | 0.9          | 1.2   | 0.9    | 0.8   | 0.9   |
| (ethene) | Mt. Abu  | Sahu and Lal (2006)                        | 0.3          | 0.7   | 0.5    | 0.5   | 0.6   |

Table C1. Comparison of modeled monthly average (for April) precursor mixing ratios (in ppbv) with observations at several stations

**Comment 4:** The comparison between MOZART and RADM-2 also hinges on an unknown in the model performance over India. I have seem a few papers on WRF from India that shows huge ( +/-1000 mts or more) differences in PBL heights by just using two different PBL schemes in the model. If MOZART is producing more ozone in the upper troposphere and is getting entrained into the PBL, where is the evaluation of PBL heights or entrainment rates in the study.

Response 4: We agree that choice of PBL scheme could affect local pollutant concentration especially over complex terrains, however Singh et al. (2016) observed little impact on surface ozone and larger impact on aerosols in this season during the Ganges Valley field campaign. The usage of the MYJ PBL scheme in this study is motivated from previous studies (Kumar et al., 2012a; Ojha et al., 2016). Nevertheless, following the reviewer's suggestion we conduct a

simulation using another parametrization (Yonsei University Scheme) and analyse its effect on our conclusions.

Comparison of monthly average (in April) planetary boundary layer heights between the two PBL schemes (Fig. C1; Fig. S8 in revised supplement) revealed that the differences are mostly within  $\pm 150$  m with Yonsei scheme generally resulting in higher PBL heights over India. Nevertheless, the chemical tendencies combined with vertical mixing tendencies of surface O3 are found to be nearly similar with Yonsei scheme (Fig. C2; Fig. S9 in revised supplement) as in the base runs using the MYJ scheme (Fig. 9b in manuscript) with MOZART still producing higher ozone aloft (not shown) as in the original runs. Thus changing the PBL scheme still results in production of more ozone aloft in MOZART which is getting mixed with near surface air showing that our conclusions are not affected. This information is provided in the revised version of manuscript (Page: 10, Lines: 374-382).

Figure C1. Difference in monthly average (in April) PBL height in meters between simulations with Yonsei and MYJ parameterization (i.e. base run) with HTAP-RADM2 setup.

**Figure C2.** Average net daytime surface ozone chemical +vertical mixing tendency (in ppbv  $h^{-1}$ ) for April during 0630-1230 IST for HTAP-RADM2 and HTAP-MOZ setupbut with the Yonsei PBL scheme.

**Comment 5:** Why is MOZART producing more ozone in the upper troposphere than RADM? Is it because the photolysis rates used in RADM different than the ones used in MOZART? I am guessing the photolysis code used for both RADM and MOZART are the same – but please check.

Response 5: Because of the way the two mechanisms RADM2 and MOZART are implemented into WRF-Chem, they use different photolysis schemes: RADM2 uses the Madronich TUV or Fast-J scheme, and MOZART uses the "Fast" TUV (Madronich F-TUV) scheme, which is based on the same physics as the Madronich TUV scheme, but designed to run faster. The differences between the two Madronich photolysis schemes are further described in the supplementary material to Mar et al. 2016.

In the present study although RADM2 uses the Fast-J photolysis scheme, a sensitivity simulation with Madronich TUV revealed similar surface ozone mixing ratios and chemical tendencies at various model levels with small differences (

**Figure C3.** Percentage difference in monthly average surface ozone (ppbv) during April between S4RS-RADM2\_kf run (using Kain-Fritsch convection scheme) and S4RS-RADM2 base run (using Grell 3D scheme).

**Comment 7:** I have also noticed that lines 130/131 probably refer to spectral nudging and not really a FDDA. Do you have or assimilated any observational meteorological data from the Indian Meteorological Department (sondes, surface weather stations etc) to perform the FDDA? Performing spectral nudging to ERA probably is not a good idea, unless you can establish that it is a good representation of synoptic scale conditions over India during this period. Many instances (specially at 12 km resolution) it is better to run the model in data poor areas with model physics than nudging the entire wind profile to ERA or any other reanalysis

Response 7: No we did not use spectral nudging. Grid analysis nudging (grid\_fdda =1) has been used to nudge the model towards the Era enterim reanalysis fields. Such nudging is shown to well represent the synoptic scale conditions over India (Kumar et al., 2012a; Ojha et al., 2016; Girach et al., 2017).

**Comment 8:** Have you evaluated the model synoptic scale meteorology for the simulation period with any observations?

Response 8: Numerous studies have shown that WRF-Chem reproduces the synoptic scale meteorology over the Indian region with sufficient quality for its use to drive chemical simulations (e. g. Kumar et al., 2012a). Further nudging towards the reanalysis fields limits the errors in simulated meteorology (e. g. Kumar et al., 2012a; Ojha et al., 2016; Girach et al., 2017). Nevertheless, we now include evaluation of model simulated water vapour, temperature and wind speed against radiosonde observations (Fig. C4; Supplementary material, Fig. S3). We also find that model simulated meteorology is in good agreement (within 1-standard deviation variability) with the observations. This is discussed in the revised version of the manuscript (Page: 6, Lines: 208-217).

---

## Author Response (AR2)

**Comments to the Author:**

The focus of this paper is an exploration of surface ozone levels predicted over India and adjacent regions using the WRF-Chem model, with two different chemical mechanisms, and three different emission inventories. The resulting predicted ozone fields are compared to each other, and to observations. The manuscript has received two detailed reviews, which raise a number of issues, the majority of which the authors have addressed, although with some areas where further refinement would improve the quality of the manuscript, and noted below.

**Response: We thank the editor for thoroughly reviewing our work and for his valuable suggestions. The points raised (normal text) are addressed here (bold text). The corresponding changes made in the manuscript are highlighted in blue colour for added text and in red colour (strike out) for removed text. We are obviously open for any further comments and improvements in the manuscript, if needed.**

**Comment:** The more substantive comment relates to the overall focus of the work. In particular, the referees challenge the use of emission inventories from very different years (2006, 2010, 2012), in a simulation for 2013, and which is compared with observational data acquired mostly between 2009 and 2013, but in some cases dating back to 2004 or earlier. The concern arises as there are substantial and well documented trends in ozone precursor emissions, and ozone levels, over this region. There are secondary concerns over the focus upon just O3 rather than NOx and VOC as the model metrics (although addressed to an extent in the revised version), and station specific dependencies in some locations which regional model resolution cannot address.

**Response: The old observations pointed out by the editor (2004 and before) has been removed in the revised version. We do realize that regional models cannot resolve the station specific dependencies of some locations. Still it should be noted that we have conducted simulations at higher resolution (12 km) than those in previous evaluation studies over South Asia (for example - 45 km by 45 km in Kumar et al., 2012b; 30 km by 30 km in Ojha et al., 2016). Also many of the stations used in the study are somewhat away from the roadside and generally inside the campus of universities/ institutes (see references provided in the Table 4) and therefore representing homogeneous conditions.**

**To make better model predictions at further higher resolution (than 12 km), development of finer resolution inventories than the ones used in the current study is also required over the region (HTAP and SEAC4RS are at 0.1 by 0.1 deg resolution which is around 11 km by 11 km). So we also recommend preparing high-resolution regional inventories for the anthropogenic emissions of O3 precursors over South Asia, also accounting for year-to-year changes. This is also briefly discussed in the revised version (Page: 15, Lines: 583-586).**

The results in the amended manuscript (i.e. most figures, tables) focus upon the difference in modelled ozone and ozone tendencies between the different inventories and chemical mechanisms, and exploration of the factors underpinning these. This is the appropriate focus for the manuscript, as the results cannot be readily compared with absolute ozone levels using ozone observations 15 years adrift from the inventory, and different meteorological years etc. However the paper text still focuses substantially upon the comparisons with measurements – this focus should be adjusted, to comparison between inventories and chemical mechanisms, in a revised manuscript. The authors response to this point raised by the external referees – that better data do not exist – may be correct, but should lead to the conclusion that the comparison cannot therefore usefully be performed (at least, without explicit consideration of precursor and ozone observational trends, local station factors, and NOx and VOC measurements).

**Thanks for the suggestions. The paper text focusing upon the comparisons with measurements have been revised by comparing the patterns in different chemical environments (Page: 9, Lines: 328-335; Page: 12, Lines: 438-439; New Supplementary Figs. S6, and S12), as discussed in the response to Referee 1, comment 1, below. Focus has been on inter-comparison of model results more now. A simulation of differing year (2010), in addition to 2013, is kept in the manuscript to show the variability that it can induce in the simulated ozone and to show that the main conclusions of the paper are not affected if we simulate for a different meteorological year.**

Referee 1 comments

**Comment 1**: Focus should be upon ozone production between models, not absolute ozone levels. The (many) uncertainties in model chemistries need to be outlined in the paper, both between the two mechanism used, and overall

**Response 1: Thanks for the suggestion and we have revised the manuscript in this direction. We include new figures (see revised supplement material; Fig. S6 and S12) to compare the ozone build up using ΔO3, which is the difference between diurnal mean and hourly values, for model simulations and observation at all stations. Thus the figure is useful in comparison between different simulations and observations without actually considering absolute ozone levels. The corresponding text is included in the revised manuscript (Page: 9, Lines: 328-335; Page: 12, Lines: 438-439).**

**The uncertainties relating to the chemical mechanisms have been added to the revised manuscript (Pages: 9-10, Lines: 355-361) as follows:**

**"Choice of chemical mechanisms in the regional models can also be an important element in the prediction of ozone. Inclusion of additional chemical species along with insufficient information on region-specific speciation factors could induce uncertainties to the predicted ozone. Further, in order to reduce the computational costs most chemical mechanisms in the models make use of lumping approach to reduce the number of chemical reactions thus avoiding treatment of all chemical species (Zaveri et al., 1999; Sarkar et al., 2016). In addition, different reaction rate constants, photolysis and dry deposition schemes used in the mechanisms are some of the factors leading to the uncertainties".**

**The differences between the chemical mechanisms used in the study also have been mentioned in the revised manuscript (Page: 4, Lines: 150-154; Page: 10, Lines: 366-370) as follows:**

**"In the present study, the photolysis rates are calculated using the Fast-J photolysis scheme (Wild et al., 2000) in RADM2 simulations and the Madronich FTUV scheme in the MOZART simulation. In WRF-Chem, the Madronich F-TUV photolysis scheme uses climatological $O_3$ and $O_2$ overhead columns. The treatment of dry deposition process also differs between RADM2 and MOZART owing to differences in Henry's Law coefficients and diffusion coefficients."**

**Comment 3**: the comparison with observations is of limited use to guide the choice of "correct" answer given the issues outlined above – better comparison would be with newer emissions data. Local surface site factors (e. g. roadside etc) will dominate the NOx levels and hence O3 levels – which cannot be captured in a regional model. Observations of "NOx" are probably (mostly) NOy, if heated Mo converter instruments are used for NO2 detection.

**Response 3: We agree and now explicitly mention these limitations in the revised version (Page: 13, Lines 490-493; Page: 15, Lines: 566-568). Many of the stations used in the study are somewhat away from the roadside and generally inside the campus of universities/ institutes (see references provided in the Table 4).**

**We agree that employing more sensitive techniques (e. g. blue light converter for $NO_2$) in future would provide better insights into model performance in reproducing NOx over India. This is also mentioned in the revised version (Page: 7, Lines: 242-243).**

**Comment 5**: please include an example of actual j values from the two photolysis schemes to allow the reader to understand the impact – e.g. a simple comparison of jNO2 and jO(1D) for a surface point in the centre of the domain at midday

**Response 5: Variation of required photolysis rates from 1000 IST to 1400 IST and their specific values at surface point in the centre of the domain (just for midday) is provided below for 15$^{th}$ April 2013. This is also now added in the revised manuscript (Page: 14, Lines: 534-535; Fig. S13 in supplement).**

[Figure]

| Chemical mechanism | Midday O1D photolysis rate (min-1) | Midday NO2 photolysis rate (min-1) |
|:---:|:---:|:---:|
| RADM2 | 0.0023 | 0.5375 |
| MOZART | 0.0023 | 0.5528 |

**Comment 11**: it would be useful to include the explanation for HCHO/NOy ratio in this manuscript. The work of Sillman et al refers primarily to a US context where VOC emissions (and hence VOC oxidation yields, ie HCHO formation) may be quite different to India – and referred to a specific trajectory duration / timepoint post emission - so the transition from NOx to VOC limitation may not occur at 0.28 in India, and will vary with post-emission processing duration.

**Response 11: Sillman (1995) evaluated the correlation between O3-NOx-VOC sensitivity predicted by photochemical model and CH2O/NOy ratio. The correlation has been derived combining results from serial computations with the model by varying the anthropogenic and biogenic emissions, and meteorology. The method has been successfully employed in investigating ozone distribution over the South Asia (Kumar et al., 2012b), East Asia (Geng et al., 2007; Tie et al., 2013), and Europe (Mar et al., 2016). Tie et al (2013) reported similarities between the results based on the CH2O/NOy ratio and those following another method described by Kleinmann et al. (2003) over Shanghai.**

**The explanation has also been added to the revised manuscript (Page: 8, Lines: 287-293).**

**Additionally, the method has been used here to differentiate the ozone sensitivity between inventories so a shift in transitional value of 0.28 would not change the broad conclusions that differences exist in ozone sensitivity to precursors among model simulations.**

**Comment 13**: The decision to turn off the aerosol radiation feedback is appropriate for the focus upon comparison between inventories (but not for comparison with absolute observed O3 levels), this should be more clearly brought out / explained in the manuscript

**Response 13: The suggested limitations are now clearly mentioned in the revised manuscript (Page: 13, Lines 490-493; Page: 15, Lines: 566-568). Comparison with absolute observed O3 levels also has been substantially reduced.**

**Comment 14:** Give specific values for the modelled vs observed water vapour data (ie comparison of means at surface / through profile, sonde profiles used etc).

**Response 14: Meteorology is nudged towards Era-interim reanalysis thus limiting errors in simulations. Additionally, meteorological inputs and setup are similar among different simulations and therefore would not lead to significant differences in simulated chemistry.**

**Specific values are being provided below at lowest level of sonde observations available.**

| Station | Pressure level (hpa) | Modeled water vapour (g/kg) | Observed water vapour (g/kg) |
|---------|---------------------|----------------------------|------------------------------|
| Delhi | 925 | 6.5±1.9 | 6.7±1.6 |
| Bhubaneshwar | 1000 | 18.7±1.4 | 18.4±3.5 |
| Ahmedabad | 1000 | 9.7±3.0 | 11.1±2.2 |

Manuscript Number: acp-2016-1083

Referee 2 comments

**Comment 1**: please advance explanations for the difference in model performance (between models, not vs observations) as this is the focus of the study and paper title

**Response 1: Manuscript has been revised to focus more on the inter-comparison of model results. As mentioned to the response to referee 1 (comment 1), we have added new figures (see revised supplement material; Fig. S6 and S12) to compare the variability / patterns using ΔO3 (mean O3-hourly O3) between model and observations, which somewhat minimizes the effect of absolute ozone levels on comparison. The corresponding text is also suitably revised (Page: 9, Lines: 328-335; Page: 12, Lines: 438-439).**

**The old station data (pre 2004) have been completely removed and the comparison with observation in the revised manuscript is mostly based on diurnal variability rather than absolute numbers. The comparison of model results with observation (revised by excluding 2004 and old observation) averaged for all sites in the manuscript is however retained as to indicate (roughly) how average ozone values compare between model and recent observations (2009-2013), with a clear mention of the limitations associated with such a comparison (Page: 13, Lines 490-493; Page: 15, Lines: 566-568).**

**Comment 2:** please expand on the impact of biomass burning effects in this work [not from literature] – ie from that aspect of the different emission inventories?

**Response 2: The anthropogenic emission inventories used in the study exclude biomass burning emissions. So we have used biomass burning emissions from Fire Inventory from NCAR (FINN) that has been kept same in all simulations (Page: 5, Lines: 185-186). FINN emissions have been used in numerous modelling studies (e.g. Kumar et al., 2012b, Amnuaylojaroen et al., 2014). As the focus of this paper is only on reporting the differences between anthropogenic emission inventories in predicting ozone, the analysis of impact of biomass burning has not been conducted. Authors are collecting observations of fire tracers using a mass spectrometer and would follow up on the suggestion in a separate study.**

**Comment 8:** the pre 2004 data may not be appropriate to use for the reasons noted above. If they are to be retained explicit consideration of the trends in O3 is needed – however the manuscript focus should compare the models.

**Response 8: Pre-2004 data is completely removed in the revised version of the manuscript in all figures (see e.g. revised Table 4, Fig. 1, Fig. 6, Fig. 11 etc).**

**Abstract** – needs to mention the second chemical mechanism, ie MOZART. Discussion of absolute model performance should be reduced given issues noted above.

**Response: Name of mechanism is now mentioned (Page: 1, Line: 30). Discussion of absolute model performance is also reduced with more discussions on variability/ diurnal patterns and ozone build up using ΔO3 now (Page: 9, Lines: 328-335; Page: 12, Lines: 438-439; New Supplementary Figs. S6 and S12).**

**References**

[revised manuscript text omitted]

---

## Author Response (AR3)

**Comment:** Thank-you for the substantial changes to the manuscript. Two minor revisions are requested. To comment clearly upon the comparison of modelled and measured NOx mixing ratios (as shown in Table S1) within the main manuscript, e.g. in sections 3.1 and 6.

**Response: We thank the editor for careful evaluation of the manuscript and constructive comments.**

**The comparison of modelled and measured NOx is discussed now within the main manuscript (section 3.1-Page: 8, Lines: 294-305; as "Some of the ozone precursors (NOx/ NOy, ethane and ethene) are also compared between model and recent measurements over few stations (Table S1). Significant differences are seen in model simulated NOx mixing ratios among different emission inventories (e. g. 6.5-30.5 ppbv at Kanpur) over the urban stations in the IGP. Model typically overestimated NOx mixing ratios at Delhi except in INTEX-RADM2 simulation, which showed an agreement with observations within 1-standard deviation. Total nitrogen oxides (NOy) showed relatively similar levels among different inventories (2.7-3.2 ppbv) at a high-altitude station (Nainital) in north India and were only slightly higher than observed mean (1.8±1.6 ppbv). In contrast with the stations in northern India, NOx levels over a rural station Udaipur in western India are underestimated by a factor of 4. On the other hand, modelled ethane mixing ratios are underpredicted by a factor of about 2 whereas modelled ethene mixing ratios agree relatively well with observed values at Nainital in INTEX-RADM2 and S4RS–RADM2 as compared to HTAP-RADM2. More in situ observations, especially of ozone precursors, may provide better insights into the performance of the numerical models and employed emission inventories over this region."**

**And in section: 6 - Page: 14, Lines: 542-545; as "Modelled levels of ozone precursors showed significant differences among simulations employing the three emission inventories with an overestimation of NOx levels at urban stations in the IGP region. Evaluation of model simulated levels of ozone precursors over a network of observations is highly desirable, as conducted for ozone in this study."**

**Comment:** Technical correction - clarify that figs S2 onwards refer to $O_3$.

**Response: We hope that the editor wishes to mention Table S2 onwards rather than Fig. S2 onwards. Thanks for pointing this out, which is clarified in revised version.**

[revised manuscript text omitted]